# Not All Maca Is Created Equal: A Review of Colors, Nutrition, Phytochemicals, and Clinical Uses

**DOI:** 10.3390/nu16040530

**Published:** 2024-02-14

**Authors:** Deanna M. Minich, Kim Ross, James Frame, Mona Fahoum, Wendy Warner, Henry O. Meissner

**Affiliations:** 1Human Nutrition and Functional Medicine, Adjunct Faculty, University of Western States, Portland, OR 97213, USA; 2Food & Spirit, LLC, Port Orchard, WA 98366, USA; 3Symphony Natural Health, Inc., West Valley City, UT 84119, USA; kim.ross@symphonynaturalhealth.com (K.R.); mona.fahoum@symphonynaturalhealth.com (M.F.); dr@wendywarnermd.com (W.W.); 4Symphony Natural Health Institute, West Valley City, UT 84119, USA; 5Kim Ross Consulting, LLC, Lakewood Ranch, FL 34211, USA; 6College of Nutrition, Sonoran University of Health Sciences, Tempe, AZ 85282, USA; 7Symphony Natural Health Holdings Inc., Craigmuir Chambers, Road Town, Tortola VG1110, (BVI), UK; james.frame@naturalhi.com; 8Natural Health International Pty Ltd., Sydney, NSW 2000, Australia; 9Meridian Medicine, Seattle, WA 98133, USA; 10Bastyr Center for Natural Health, Bastyr University, Kenmore, WA 98028, USA; 11Wendy Warner, MD, PC, Yardley, PA 19067, USA; 12National Institute of Complementary Medicine, Health Research Institute, Western Sydney University, Building J, 158-160 Hawkesbury Road, Westmead, NSW 2145, Australia; dr.meissner@ttdintnl.com.au; 13Therapeutic Research, TTD International Pty Ltd., 39 Leopard Ave., Elanora-Gold Coast, QLD 4221, Australia

**Keywords:** adaptogen, glucosinolates, hormones, *Lepidium meyenii*, Lepidium peruvianum, maca, menopause, phenotype, prostate, reproductive health

## Abstract

Maca (*Lepidium meyenii*, *Lepidium peruvianum*) is part of the *Brassicaceae* family and grows at high altitudes in the Peruvian Andes mountain range (3500–5000 m). Historically, it has been used as a nutrient-dense food and for its medicinal properties, primarily in enhancing energy and fertility. Scientific research has validated these traditional uses and other clinical applications by elucidating maca’s mechanisms of action, nutrition, and phytochemical content. However, research over the last twenty years has identified up to seventeen different colors (phenotypes) of maca. The color, hypocotyl size, growing location, cultivation, and post-harvest processing methods can have a significant effect on the nutrition content, phytochemical profile, and clinical application. Yet, research differentiating the colors of maca and clinical applications remains limited. In this review, research on the nutrition, phytochemicals, and various colors of maca, including black, red, yellow (predominant colors), purple, gray (lesser-known colors), and any combination of colors, including proprietary formulations, will be discussed based on available preclinical and clinical trials. The gaps, deficiencies, and conflicts in the studies will be detailed, along with quality, safety, and efficacy criteria, highlighting the need for future research to specify all these factors of the maca used in publications.

## 1. Introduction

Maca refers to two distinct species known as *Lepidium meyenii* and *Lepidium peruvianum* [1], which are classified as the wild and cultivated forms of the plant, respectively. Maca is an annual cruciferous root vegetable and one of 249 known *Lepidium* species of plants [2]. It belongs to the same botanical *Brassicaceae* family as the turnip, cabbage, mustard, and broccoli, yet it is phytochemically distinct from this vegetable group [3]. Its predominant and native growing location is 3500–5000 m above sea level in Peru’s high, harsh-weathered Andean plateaus [4,5,6,7,8].

Due to its rising consumer demand in China, it has also begun to be cultivated in select areas with high altitudes, such as the Yunnan province of China (2800–3500 m) [9,10] and even Tibet (above 3000 m) [11,12]. However, the maca cultivated in these non-native locations exhibit different characteristics since the growing environment can significantly affect the plant phenotype and composition [8]. As reported by the American Botanical Council, there is a potential toxic feature of Chinese maca due to the use of pesticides and herbicides to accommodate for the difference in altitude relative to the Andean highlands, alongside chemical contamination in the agricultural settings in parts of China, especially the Yunnan province [13]. In 2018, the Botanicals Adulterants Prevention Program reported on maca powder being diluted with corn, wheat, and yam powders [14]. While there was an initial boom in the production and prices of Chinese maca, the demand and prices have declined drastically over the years [15]. Peruvian maca continues to uphold its superiority, as is seen by its relatively higher market prices [15]. Compared with Chinese maca, it is considered distinct in appearance and more pungent in aroma and taste [15] (Figure 1).

Depending on how the maca is processed and prepared, there can be significant alterations in parameters such as safety, stability, quality, bioactives, and clinical efficacy [8]. Exposure to environmental stressors, like altitude, ultraviolet radiation, sunlight, temperature extremes, intense wind, and varying moisture conditions, as well as soil, minerals, and microbiomes, all contribute to maca’s complex, therapeutic phytochemical profile [3,17], a concept referred to as xenohormesis [18]. Thus, these factors must be considered when evaluating maca in research studies and may even be implicated in disparate findings.

Maca, first described in 1553, serves as a dietary staple of native Peruvians, particularly in its dried hypocotyl (tuber) format at >20 g daily [3]. As a therapeutic agent, it has been used in many preparations primarily for energy, fertility, libido, and as a vitality tonic for aging [3,19,20]. Over the years, the research citations listed for the search term “maca” through the National Library of Medicine PubMed^®^ database started with just two references in 1961. As of November 7, 2023, using the search terms “maca,” “maca” [in title], “maca” [in title/abstract], “Lepidium meyenii” [in title/abstract], and “Lepidium peruvianum” [in title/abstract], results in 843, 256, 528, 292, and 17 findings, respectively [21]. Beginning with the early 2000s until the present day (2023), preclinical and clinical research has provided data that would expand maca’s use into other areas of health such as menstrual cycle regulation [22,23], menopausal symptoms [24,25,26,27], osteoporosis [28], sperm quality [29,30,31,32,33,34,35], memory [36,37,38,39], mood [36,40], prostate health [41,42,43,44,45,46], and fitness optimization (e.g., reducing inflammation and increasing strength) [47,48,49]. Even though maca has broader applications, there has been historical research emphasis on its ability to modify the endocrine system, pioneered by the work of Gonzales et al. on males [29,31,33,42,50,51] and Meissner et al. on pre- and post-menopausal women [24,25,26,27]. Meissner et al. continue to conduct ongoing research at five universities in Poland to investigate the use of different maca phenotypes to treat specific medical conditions associated with various menopausal symptoms, men’s health, and even prevalent health areas of concern shared by both genders [52].

Over the past two decades, alongside its increasing presence in scientific publications, maca has become popular in the supplement industry as capsules, powders, gummies, tablets, liquids, and tinctures. The rise in consumer-friendly formats has risen in parallel with interest in a non-pharmaceutical, low-cost, non-toxic drug option for various conditions. Most notably, maca has been colloquially referred to as “Peruvian Viagra” as a substitute for medications typically used for erectile dysfunction, like sildenafil and tadalafil [8,53,54]. In addition to sexual health, the interest in maca as a viable alternative to medications has been more extensively explored for benign prostatic hyperplasia (BPH) and menopausal symptoms due to potential contraindications or side effects that can occur with drugs like finasteride [55,56,57] or even hormone replacement therapy (HRT) [58]. Some women have explored maca as another option or even in conjunction with in vitro fertilization (IVF). In a survey of 95 women seeking infertility treatment, maca was on the list of herbal supplements commonly used [59]. Of those participants taking maca, 71.4% took it for fertility, and 28.6% used it for general health and wellness [59].

In the modern day, maca has been increasingly visible as part of nutritional regimens in the format of a powdered “superfood” [60] available in grocery stores, without much distinction or delineation as to the specific type (color or phenotype) of maca the product contains. An online search using the term “maca” on Amazon.com under the subcategory of “vitamins, minerals, and supplements” provides a listing of over 3000 results [61]. Furthermore, maca dietary supplements do not typically list standardized compounds to ensure reliable results related to known actives [62]. Therefore, there can plausibly be mixed, inconsistent physiological responses to maca powder due to this point being largely unknown. A few companies in the food and herb sector have begun to feature different colors of raw and gelatinized maca powders or tinctures (yellow, red, and black) for designated health functions, although this effort has been minimal [63,64]. Due to the limited and sometimes conflicting research on health applications based on phenotype, the recommendations of which color to use are, in some cases, debatable and may be, in other cases, potentially incorrect.

To date, maca colors and sources have been grossly overlooked in the methodology of published research studies. However, with the resurgence of interest in maca, there is an emerging groundswell of exploration into the types of maca, which can be differentiated by color, corresponding phytochemicals, and biological effects [65]. The scientific literature provides more significant recognition of the importance of knowing maca’s color, location, and format to determine its physiological functionality. The main objectives of this review are to compile research on the known colors of maca for which there are data, provide a summary of research on those colors, and evaluate the discrepancies, which have not been conducted until now.

## 2. Species of Maca

The species of maca are distinct in appearance and phytochemical profiles, with *Lepidium meyenii* Walpers being the wildcrafted form of Peruvian maca first described by German botanist Gerhard Walpers in 1843 [8] and *Lepidium peruvianium* Chacón de Popovici, the cultivated, domesticated version of maca, identified by Dr. Gloria Chacón de Popovici, in which the colors were characterized [1,66]. There has been some taxonomic debate over the years about these two species, although a more detailed analysis suggests differences in physical appearance, phytochemicals, and DNA [8,67,68]. Along these lines, many scientific publications and even commercial natural products list *Lepidium meyenii,* which can include wild maca from a variety of countries such as Bolivia, Colombia, Brazil, and Argentina; however, it may be that what is used is *Lepidium peruvianium* [1]. While these two species’ names have been referred to as synonyms, distinctions exist [1]. Meissner et al. mentioned that much of the research has been conducted on the cultivated Peruvian maca (*Lepidium peruvianum*), though it is referred to as the wildcrafted maca (*Lepidium meyenii*) within the scientific literature [24]. There is an ongoing discussion about making a finalized decision on United States Pharmacopeia (USP) monographs for maca root, maca root powder, and maca root glucosinolates dry extract based on validated methods of analysis [69].

## 3. The Significant Role of Environmental Factors and Processing Methods

### 3.1. Environmental Factors

The location (altitude, soil, and climate) the maca is grown within can have a significant impact on the quality and active ingredient profile of maca, with Chinese maca (typically grown in the Yunnan and Pamirs regions of China) [9,70] being different from native Peruvian (Junín and Ancash regions) maca [71]. Even within Peru, there are phytochemical differences between maca sourced from different altitudes and the two primary locations where it is grown [71]. Notably, when maca is cultivated at altitudes under 3500 m, one of its phytochemical classes, macamides, is reduced [65]. Geng et al. used mass spectral fingerprinting, metabolomic analysis, and genetic sequencing to assess 71 maca samples from Peru and China, 39 commercial maca supplements from 11 different companies, 31 unprocessed maca tubers, and a maca non-tuber historical sample [68]. There were compositional differences between the maca samples originating from these two different countries of origin as well as the color [68]. Altitude is a significant variable resulting in different colors (purple and red) and correspondingly higher phytochemicals like glucosinolates not seen in other colors [72].

Further, researchers have explored how the cultivation site and growth conditions, including variables such as soil and climate, impact the secondary metabolites of maca. Zhao et al. collected samples of yellow, violet, lead (gray), and pink maca from two locations in Peru on land that had never been cultivated for maca and land that had been previously cultivated two to three years earlier [73]. Significant differences in the metabolites existed, leaving the authors to conclude that the planting site, and all of its accompanying aspects such as the soil, microclimate, and hydration conditions, is the “major determining factor” in the concentration of metabolites in maca, perhaps even more than the actual color. Similarly, another study utilizing quantitative analysis concluded that geographical origin, rather than color, had a more critical role in the macamide content of maca [74]. Along these lines, maca has a characteristic odor, which may be traced to the point of origin [75].

Maca is often planted in the same area in rotation with other crops, although expansion has been needed into other regions within Peru due to increasing demand [65,76]. As a result of repeat planting, there can be changes in the composition of the soil, the soil microbes, and even the maca phytomicrobiome, which could each conceivably alter the quality of maca and its sensory properties such as flavor [65,77]. Evaluation of maca from newly cultivated terrains suggests improved sensory quality [65]. Therefore, best practices are to allow for crop rotation over 4–10 years or more [65]. Traditional Andean farming practices recommend crop rotation for up to seven years to help reduce pests [78]. Commercial plantations are rotated on an average of three years due to the depletion of nutrients in the soil [52].

Another point to consider with the use of the maca tuber is its ability for the root of a plant to accumulate heavy metals from the soil where it is grown. It has been documented that maca can contain toxic metals due to soil contamination from the mining industry in Peru [79] or potentially from naturally high levels of heavy metals due to prehistoric lava flows and soil formation [80]. Specifically, it may be grown in soil with relatively higher concentrations of metals like nickel [81], arsenic, cadmium, and lead [79]. One study analyzed yellow and purple maca for toxic metals and essential minerals and concluded that cadmium and lead concentrations in both the soil and hypocotyls exceeded permissible limits [79]. Zinc levels were also elevated in these types of maca. These findings suggest that soil remediation may be needed to minimize toxic metal intake [79]. However, more experienced farmers and manufacturers of maca have long implemented a practice of assessing land intended for cultivation on a plot basis by pre-testing soil for heavy metals [52]. Hence, it is necessary to ensure proper quality-control measures, such as toxic metal analysis, on all maca cultivated on a lot basis and evaluate how each maca color may be subject to differences in their bioaccumulation [79].

### 3.2. Processing Methods

The part of maca used and its growing stage will determine the contents of the end product [82]. The edible portions are the hypocotyl (tuber) and tap root, while the leaves, flowers, stems, and seeds (contained in a silicle) are less utilized [83,84] (Figure 1). The size and weight of the hypocotyls can vary [67], and it is in this part of the plant where colorful pigments such as carotenoids and anthocyanins are contained in the skin and sometimes the flesh [65]. Generally, hypocotyls are the plant part of maca harvested and processed for food and supplements, rather than the leaves, which are sometimes used for animal feed [65]. The hypocotyls are higher in glucosinolates, macaenes, and macamides, and the leaves are noted to have greater beta-sitosterol and total phenols [65]. The yellow color is typically associated with carotenoids or anthocyanin intermediates, whereas the reddish-purple hue is often due to anthocyanins [6,65,85], yet the analyses for these phytochemicals in each color of maca are lacking. Some anthocyanin compounds would be inherently susceptible to degradation with exposure to heat or water due to their water-soluble nature [85]. In contrast, carotenoids tend to be fat-soluble and more resistant to the effects of heat [86].

Often, maca hypocotyls are subject to gelatinization due to their relatively high starch content [7]. Gelatinization involves an extrusion process incorporating short-term high pressure, temperature, and moisture after drying and pulverization into a powder [87]. The result is a more digestible end product with bioavailable actives [24,88], which may be favorable for select populations such as those following a diet low in fermentable oligosaccharides, disaccharides, monosaccharides, and polyols (FODMAPs) [89] or those with digestive issues. In addition to digestibility, gelatinization may help to reduce the content of goitrogens, a class of compounds commonly found in a variety of plants, including legumes, cruciferous vegetables, and maca that are known to interfere with thyroid hormone activity [90,91]. Goitrogens can be inactivated through methods that involve moisture and heat [90,91]. While gelatinization has benefits, it can also result in a more hygroscopic product that is less stable [92].

Moderate pressure within a specific temperature range would provide ideal conditions for myrosinase, the enzyme that can degrade glucosinolates into its many metabolites [93,94]. Raw maca may be about 20% higher in glucosinolates compared with the gelatinized format [95], and storage of maca tubers in the Andean highlands for up to seven years has not been known to result in significant losses of glucosinolates (only a 9–12% loss) [52]. Therefore, manufacturing the product closer to when it is required would be a better practice due to the balance of the raw versus gelatinization issues that can arise. Manufacturing practices for maca products need to be attentive to these details to ensure a more efficacious, reproducible product. Moreover, each color of maca may require specific handling based on the actives they are known to contain.

It is worthwhile to consider the effect of storage on the microbial contamination of maca hypocotyls. Meissner et al. [67] evaluated the microbial contamination of four colors of maca hypocotyls. Yellow hypocotyls had the highest Gram-positive aerobic *Bacillus* strains, whereas the black maca had no detectable or low microbial levels of Gram-positive cocci strain colonies [67]. Therefore, due to compositional differences, there may be storage requirements for each color of maca to maintain its microbial integrity over time.

Finally, the post-harvesting practices (e.g., mechanical breakdown, fermentation, drying) of maca also impact its active constituents [96]. The traditional drying methods have been documented to be superior to oven-drying in commercial operations [87]. The traditional open-field drying methods take about two to three months and include the climate changes that naturally occur in the Andes mountains, such as freeze–thaw cycles and intense ultraviolet (UV) radiation [94,96]. Compared to fresh maca hypocotyls, this process can alter the profiles in glucosinolates, free fatty acids, and amides [72,94,96]. While some of these practices have been detailed in the scientific literature, there is a lack of research identifying the best procedures to optimize actives in each color of maca.

To summarize, the cultivation location, the size and weight of tubers, and their anti-bacterial activity, including the soil compositional difference where the maca is grown, along with harvesting and drying methods, are all factors that may be responsible for differences in the therapeutic functionality of maca phenotypes used in maca-containing formulations.

## 4. The Nutrition of Maca

As a fresh root, maca contains more than 80% water [17]. Peruvians commonly drink it as a juice at home and when compared against non-consumers of maca, experience several health benefits, including higher levels of estradiol and testosterone, lower systolic blood pressure, and serum levels of interleukin-6 (IL-6), as well as an improvement in chronic mountain sickness and better performance on a lower-limb strength test [97]. Indeed, these benefits would seem to come from consistent consumption in the daily diet over years, if not decades, when considering that published clinical research over short periods (weeks or even months) from raw, gelatinized, or even extracts of maca have not demonstrated statistically significant efforts on hormones. There can be variability in maca’s nutrient levels depending on several factors: the location grown [7,72], its size and weight [67], the part of maca tested [60], and postharvest conditions involving drying, storage, and maceration [83]. Maca’s macronutrient composition as a dehydrated powder includes carbohydrates (55–73%, mostly starch), fiber (8.2–25.6%), protein (8.9–21%), and fat (0.6–2.2%) [7,10,17]. Amino acids identified in various parts of maca include aspartic acid, glutamic acid, serine, glycine, cysteine, alanine, arginine, tyrosine, hydroxyproline, proline, histidine, threonine, phenylalanine, D-phenylalanine, valine, methionine, isoleucine, leucine, lysine, and tryptophan [60].

The predominant starchy nature of the maca tuber resembles the high carbohydrate content (70–85%) in other root vegetables such as potato, sweet potato, and cassava, as well as grains like wheat and maize [7]. This starch content in maca is relatively easy to gelatinize, altering its solubility and making it less shelf-stable [7]. Zhang et al. [7] tested yellow, purple, and black maca (the methods section stated the “roots” were tested) and reported no significant difference in amylose contents between them, with the range being 21.0–21.3%. However, each color of maca tested did exhibit differential viscosity, with yellow maca showing high viscosity relative to the purple and black formats, denoting that they may have defined applications in the food industry [7]. Some of these polysaccharides are being tested in vitro for their immunomodulating ability in cancer cells [98,99].

Maca is micronutrient dense, containing vitamins and minerals, such as vitamin A, vitamin B2, vitamin B6, boron, calcium, chromium, copper, iron, magnesium, manganese, niacin, sodium, potassium, and zinc [10,19,60,67,79,81,100]. The origin of maca may impact the micronutrient density. For example, one study [81] found that select phenotypes grown in a particular area of China had a wide range of sodium content (less than 30 to 2600 mg/kg dry weight), whereas Peruvian maca had a lower sodium concentration range (110–190 mg/kg dry weight). Conversely, another study [10] showed negligible differences in sodium content in maca from different locations.

Researchers at Jinzhou Medical University have suggested that essential oils, lipids, and polysaccharides are the biologically active constituents in *L. meyenii* and may contribute to its antioxidant activity, with the essential oils being the most potent contributor to free-radical scavenging action [9]. There may be more complexity to the constituents of maca. Carvalho and Ribeiro [19] reported on 101 bioactive phytochemicals in *L. meyenii* extracts. These structurally diverse secondary metabolite compounds encompass a wide range of secondary phytometabolites such as glucosinolates, isothiocyanates, flavonols, phytosterols, polysaccharides, fatty acid derivatives (2-oxononadecanoic acid, anandamide, oleamide), and alkaloids such as macaenes, macamides, and thiohydantoins [19,101]. Meissner et al. list seven categories of phytochemicals within *L. peruvianum* with known physiological relevance: amides, carbolines, catechins, cyanogenic compounds, fatty acids, glucosinolates, and imidazole alkaloids [72].

### Nutritional Differences between Colors of Maca

Researchers quantified 79 different nutrients and metabolites in three colors of Chinese maca (yellow, black, and violet) [102]. In this study, yellow maca was found to be high in carbohydrates, black maca was rich in protein, and violet maca was highest in antioxidant capacity. While the overall macronutrient (protein, carbohydrate, fat, fiber) composition between the four main maca phenotypes (black, purple, red, and yellow) was within relatively similar ranges for the Peruvian maca analyzed, Meissner et al. identified one distinct difference in fatty acid composition [67]. Namely, the fatty acid, C18:1n-9 (elaidic acid), the trans isomer of oleic acid, is substantially higher in the yellow phenotype, although the implications of that higher concentration are unknown [67].

Another study indicated that red maca had higher protein and potassium content but less soluble reducing sugars, riboflavin, and iron than black maca [9]. Analytical attempts have been made to evaluate the phytochemical content of the different colorful varieties of *L. meyenii* [9,71] and *L. peruvianum* [39,67,72,103]. Seven compounds were reliably detected in yellow, black, white, and purple maca (*L. meyenii*) samples grown in China: *p*-hydroxybenzylglucosinolate, benzylglucosinolate, *N*-benzyl-9Z,12Z,15Z-octadecatrienamide, *N*-benzyl-9Z,12Z-octadecadienamide, *N*-(3-methoxybenzyl)-hexadecanamide, *N*-benzyl-hexadecanamide, and *N*-benzyl-9Z-octadecenamide [71]. Of these samples tested, yellow and black maca were highest in glucosinolates (1.55%), followed by white (0.93%) and purple (0.76%) maca [71]. Macamides were lowest in black maca (0.15%) compared with the other colors (0.23–0.29%) [71].

The quantity of certain phytochemicals may vary with the growing stage, as demonstrated in a study assessing the transcriptomics of black maca [82]. The secondary metabolites are presumed to be responsible for the purported health benefits of maca alone or in combination. As noted above, distinct differences have been observed between phenotypes, phytochemicals, and cultivation location [71], suggesting the genes’ ability to express uniquely in those respective environments [102].

## 5. Select Maca Phytochemicals

A study in 2017 reported that among the 15 compounds found in maca, glucosinolates, alkaloids, and macaenes are the major phytochemical constituents, with glucosinolates reported as the highest compared to the others [104]. These phytochemical compounds will be explained below (Figure 2), as well as other relevant classes of plant compounds found in maca: phytosterols and thiohydantoins.

### 5.1. Glucosinolates

No single constituent has been identified for the health benefits of maca. However, significant research has been conducted on the glucosinolate fraction, most likely because of its occurrence throughout different phenotypes of maca, such as yellow, red, purple, and black varieties [19,72]. Although there is a noticeable color distinction by sight, multiple studies have suggested that the varying glucosinolates in maca hypocotyls, such as those seen with the distinct phenotypes and at specific altitudes, may be identifiers for the individual maca phenotypes [39,68,72,103].

Glucosinolates, of which more than 150 types were identified [111], are proposed to be some of the most active compounds in maca phenotypes [112]. Their significance has been recognized as a marker of quality control processes in the dietary supplement industry and as the proposed precursors to other maca phytocompounds, such as macamides, macaenes, and thiohydantoins [62]. These pungent-tasting, nitrogen- and sulfur-containing compounds and their metabolites serve several functions, such as anti-fungal, antimicrobial, and chemoprotective actions [95,113]. It is well known that glucosinolates, such as those in the cruciferous vegetable family, play a role in the metabolic detoxification of hormones and environmental toxicants [114,115]. Therefore, they may be pivotal in reducing cancer risk [116]. Additionally, they have been identified as an area of research for neurological, musculoskeletal, and cardiovascular conditions [117].

Even though they are members of the same botanical family, maca and cruciferous vegetables vary in their glucosinolate content. One analysis reported that the absolute content of glucosinolates was about 100 times higher in fresh maca hypocotyls than in other cruciferous crops [118], making this plant a desirable option for researching glucosinolate benefits. Glucosinolate content can vary extensively depending on the plant part studied. In one study using *L. peruvianum* samples obtained from an open market in Peru (dry hypocotyls) and maca plants grown at the University of California at Davis, researchers analyzed nine glucosinolates [118]. They reported the highest amount found in the following order, from highest to lowest: seeds (69.45 µmol/g), fresh hypocotyls (25.66 µmol/g), sprouts (18.5 µmol/g), dry hypocotyl (4.45 µmol/g), and leaf (3.77 µmol/g) [118]. As discussed, one of the reasons for the variation is due to the liberation of the enzyme, myrosinase, from cells in the plant when damaged through drying or mechanical sheer stress, resulting in the breakdown of glucosinolates into isothiocyanate, nitrile, and thiocyanate [17,118]. It is worth mentioning that glucosinolate (glucobrassicin) metabolites like indole-3-carbinol (commonly referred to as I3C), diindolylmethane (known as DIM), and sulforaphane are mainly found in the cruciferous vegetables [119] but are not found in appreciable levels in maca tubers. Therefore, it may be clinically beneficial to consider the therapeutic administration of both cruciferous vegetables and maca root for the different detoxification compounds they contain.

It has been suggested that glucosinolate content is a marker of quality and may reflect drying methods [96]. Levels of glucosinolates in fresh and dry hypocotyls from different regions in Peru for black, purple, red, and yellow phenotypes of *L. peruvianum* were reported by Meissner et al. [103]. The fresh hypocotyls were generally higher in glucosinolates than in the dry format. The dry hypocotyls from Junín were higher in glucosinolates than those from Ancash. The red phenotype in fresh hypocotyl and dry hypocotyl (from Junín) forms was relatively greater than the other three of the four phenotypes tested. However, the purple phenotype from dry hypocotyls originating from Ancash was higher than the other three phenotypes. In one study [68], maca from China was reported to have higher glucosinolate concentrations than Peruvian samples, while another study found glucosinolate levels higher in Peruvian maca compared to Chinese maca [104]. These findings highlight the need for more detailed specifics around the source, size of hypocotyl, drying methods, and manufacturing practices, to name a few, to ensure that similar maca samples are being compared across studies.

Nine distinct glucosinolates have been identified in a 50% ethanolic extract of black maca (*L. peruvianum*) [19,39]. One of the predominant classes of glucosinolates in this maca extract is the aromatic glucosinolates, notably benzyl glucosinolate (glucotropaeolin), accounting for a majority (up to 80%) of all glucosinolates [19,120]. Newer cell research by Tarabasz et al. suggests that specific glucosinolates may be the chemical feature of *L. peruvianum* that provides acetylcholinesterase inhibition activity, implying that the phenotypes with these marker compounds may ultimately have a role in therapeutic approaches to memory loss [39].

### 5.2. Alkaloids

In the 1960s, Dr. Gloria Chacón identified four alkaloids responsible for the fertility-enhancing effects of *L. peruvianum* [87]. The bitter-tasting, nitrogen-containing, and potentially toxic alkaloids are part of the plant defense system. Several types of alkaloids have been identified in *L. meyenii* extracts, including, but not limited to, alkaloid amides (macamides), hydantoin derivatives (macahydantoins, meyeniihydantoins, and macathiohydantoins), hexahydroimidazothiazole derivatives (meyeniins), imidazole alkaloids known as lepidiline A, B, C, and D [121,122,123], and pyrrole alkaloids referred to as macapyrrolins [19,123]. Preliminary cytotoxicity activity has been determined in some of these compounds in specific cell lines [124] and potential anti-inflammatory, anti-allergic, and anti-thrombotic effects, depending on the extract used [123].

Geng et al. [68] and Zhou et al. [104] reported that Peruvian maca (*L. peruvianum*) had higher alkaloid concentrations than maca cultivated in China.

### 5.3. Macamides and Macaenes

Macamides and macaenes are benzylated alkamides, which are natural compounds often used to identify maca accurately [39,125]. However, macamides are considered unique to maca [47], whereas macaenes can be found in other plants such as tomatoes and eggplants [125]. A review of analytical research using liquid chromatography coupled to high-resolution mass spectrometry (LC-HRMS) for *L. meyenii* extracts reported three macaenes (*C*-18 fatty acid derivatives) and twenty-three macamides (fatty acid amides) [19]. A slightly older study using different analytical methods (UHPLC-ESI-Orbitrap MS and UHPLC-ESI-QqQ MS) on a methanolic extract of maca cited only eleven macaenes [104]. Macamides are a unique class of maca phytochemicals, with more than thirty compounds identified and explored for their anti-fatigue effects [9,47]. There is a suggestion in the scientific literature that macaene and macamide compounds could be (partly) responsible for the bioactive effects of maca on sexual performance in male animals [17,126]. Macamides are also important anti-inflammatory, antioxidant, and anti-cancer compounds [125], contributing to maca’s anti-fatigue, neuroprotection, and fertility-enhancing effects [104]. In mice, macamide administration helped reduce pro-inflammatory factors and reactive oxygen species after a swim test, improving markers of grip strength and extending time spent in activity [127]. A recent study notes that the macamides in maca exhibit nuclear factor erythroid 2-related factor (Nrf2) activation [128]. Macaenes may play a role in lipid metabolism [125].

Newer research on the science of maca phytochemicals continues, with some recent publications looking at the immunomodulatory [98] and neuroprotective [129] aspects of polysaccharides from maca, along with testing isolated compounds like a natural fatty amide known as *N*-benzylhexadecanamide for its ability to enhance testosterone production [130]. Using proteomics and lipomics methods, the neuroprotective effects imparted by *N*-benzylhexadecanamide may be attributed to its effects on sphingolipid metabolism and mitochondrial function, suggesting that it could continue to be investigated for neurodegenerative diseases [131]. Comparatively, this fatty acid amide is found in higher amounts in Peruvian maca than Chinese maca [101], suggesting that elevation or other factors could be responsible for its synthesis. Macamides are thought to influence mood states by modulating the endocannabinoid system and reducing the degradation of anandamide, which acts on the cannabinoid 1 (CB1) receptor [132,133].

### 5.4. Phytosterols (Plant Sterols)

Maca also contains phytosterols such as stigmasterol, beta-sitosterol, avenasterol, campesterol, and brassicasterol, which are structurally and biochemically related to cholesterol and steroid hormones such as estrogen, testosterone, and progesterone [3,19]. Of three plant sterols measured in a gelatinized combination of maca color types known as Maca-GO^®^, beta-sitosterol was at the highest level, followed by campesterol and stigmasterol [24]. It has been suggested that these hormone-like compounds reduce plasma cholesterol levels [134] and support endogenous hormone production appropriate to the age and gender of the person [24,135]. However, the source, amount, ratio, format, and addition of other compounds may be relevant for the effects on hormone synthesis. No changes were observed in sex hormones in men or women taking plant sterols in a margarine spread [136,137].

### 5.5. Thiohydantins

A lesser-known and not as widely researched class of phytochemicals within maca is the sulfur- and nitrogen-containing thiohydantins, of which seventeen types have been identified [101]. This moiety has been claimed to have anticarcinogenic, antimutagenic, antithyroidal, and hypolipidemic properties [101].

## 6. Functions of Maca

In some manner, maca has a variety of functions, as substantiated by preclinical and clinical trials over the past decades, that include antioxidant activity, neuroprotection, antidepressant effects, immunomodulation, antimicrobial and antiviral action, hepatoprotection, glycemic reduction, UV-protection, anticancer properties, antiproliferative function, detoxification support, anti-osteoporotic activity, stress tolerance, and modulation of the hypothalamus–pituitary gland endocrine signaling [4,9,39,79,83,138,139,140,141,142]. Some of these functions have been observed with specific colors or combinations of maca colors, as detailed below. In contrast, other functions, like antimicrobial, antiviral, chemopreventative, and immunomodulatory activities, have not been effectively assigned to the color of maca due to a lack of mention in the published literature.

As a result of these properties, preclinical (animal) and clinical studies combined, maca has become known for its therapeutic effect in myriad conditions, including fertility and reproductive health in men and women [17,31,59], prostate health [44,45,51], sexual performance [17,143] and sexual desire [50], cognitive impairment and memory loss [39,144], menopause [145,146], low chronic mountain sickness scores [97], skin health [147], anemia [145], cancer [17,95,140,148], vitality [17], gastrointestinal motility [149], and osteoporosis [17,28] (Table 1). Although not confirmed, it has been suggested that the effects on mood and cognition may be due to the possibility of maca metabolites crossing the blood–brain barrier [39]. While an interesting hypothesis, much of whether maca compounds can cross this highly selective barrier may depend on specific fractions of maca used (e.g., more hydrophobic fractions rather than hydrophilic ones).

## 7. Clinical Application: Endocrine System Support

While there are numerous proclaimed traditional uses and preclinical studies on maca, this paper will highlight its role in the endocrine system. The rationale for this focus is that this area is where its central claims for use (e.g., energy, fertility, menopausal symptoms, prostate health, reproductive function) reside from a mechanistic point of view and where there exists most of the research, relative to its effects on other parts of the body.

### 7.1. Adrenal Health

In botanical medicine, maca is classified as an adaptogen [3,145]. Adaptogenic herbs are unique from other substances in their ability to modulate hormones and the immune system, assisting with maintaining optimal homeostasis. Adaptogens have a modulating effect on the body and can either tone down the activity of hyper-functioning systems or strengthen the activity of hypo-functioning systems [150]. While the primary proposed mechanism of action for adaptogens is the impact on the hypothalamic–pituitary–adrenal [151] axis, especially cortisol output, they can also display anti-inflammatory and antioxidant effects and influence gene regulation involved in modulating pathways of detoxification and stress regulation [150].

Indeed, maca’s function would fit the definition of an adaptogen; however, as discussed herein, could maca be in a category by itself? It may be separate from being solely classified as an adaptogen due to the various phenotypes having different physiological effects and modulating other systems beyond the adrenal glands. The research utilizing standardized, quality-controlled maca formulations indicates that the entire endocrine axis, consisting of the hypothalamus, pituitary gland, thyroid gland, adrenal gland, and gonads (HPTAG axis), is impacted.

### 7.2. Ovarian Health

Scientific research suggests that when combining the purported benefits of select colors, maca’s mechanisms surpass that of a classically defined adaptogen [27]. Importantly, research conducted more than fifty years ago by Dr. Gloria Chacón de Popovici [152] touted that alkaloids in maca were stimulating the testes and ovaries of rats, extending its beneficial effects past the hypothalamus–pituitary–adrenal [151] axis into the hypothalamus–pituitary–adrenal–gonad (HPAG) axis [27]. However, to date, only one select formulation of specific concentrated maca phenotypes (referred to in research as Maca-GO^®^, commercially as Femmenessence^®^) has exhibited modulation of the hypothalamic–pituitary–ovarian (HPO) axis in early postmenopausal women via clinical changes in estradiol (E2), progesterone (P), follicle-stimulating hormone (FSH), and luteinizing hormone (LH) [24,25,26,27].

Conversely, in another clinical trial, postmenopausal Chinese women from Hong Kong taking 3.3 g daily of an unspecified color of maca known commercially as Maca Power (Healthychoices, Murwillumbah, NSW, Australia) for six weeks did not experience any changes in E2, FSH, or LH, compared with six weeks on a placebo [153]. It was not stated in this study whether the maca used was raw or gelatinized. Similar to these latter findings, the researchers found no differences in these hormones in fourteen postmenopausal women taking 3.5 g of Maca Power (Incan Food, Murwillumbah, NSW, Australia) daily for six weeks with a six-week crossover on placebo. There were no significant differences in serum concentrations of these hormones (E2, FSH, LH), even though symptoms on the Greene Climacteric Scale, such as anxiety, depression, and sexual dysfunction, were improved using Maca Power [154]. A study in healthy, perimenopausal Japanese women given a maca extract standardized to at least 1.2% benzyl glucosinolate, known as Maca-BG1.2^TM^ (CPX PERU S.A.C., not currently sold commercially), for eight weeks found a 2.2-fold increase in E2 levels, although this was not statistically significant [155]. These findings, in conjunction with the Maca-GO^®^ results, highlight how important phenotype and concentration are in eliciting statistically significant effects on hormones in peri- and post-menopausal women.

Concerning menopausal symptoms, one of the studies using Maca Power mentioned above reported non-significant menopausal symptom reduction compared with a placebo (using Greene’s Climacteric Scale, the total score after six weeks of maca supplementation was 17.6 ± 10.0, *p* = 0.07, and the placebo group was 16.8 ± 9.1, *p* < 0.05) [153]. Alternatively, the clinical trials using Maca-GO^®^ reported highly significant reductions in menopausal symptoms in early postmenopausal women, notably hot flushes and night sweats, and overall symptoms, including nervousness, depression, sleep, and heart palpitations, using Kupperman’s Menopausal Index and Greene’s Menopausal Score (*p* < 0.001) [24,25,26]. These results further validate how important phenotype and concentration are for ensuring optimal clinical outcomes.

Eighty-five percent of peri- and postmenopausal women experience various menopausal symptoms, including hot flushes, night sweats, sleep disturbances, mood imbalances, loss of libido, weight gain, and vaginal dryness, that can last for decades [156,157]. Apart from debilitating symptoms impacting the quality of life, the significant loss of hormones experienced at these stages of life results in substantial increases in morbidity and life-threatening conditions, ranging from osteoporosis to cardiovascular disease [158]. These serious health implications have naturally given rise to the use of HRT and bioidentical hormone therapies (BHRT). While the introduction of exogenous hormones is a logical solution, the fluctuations of hormones experienced during perimenopause and the simultaneous increase or proportional increase, such as between E2 and testosterone (T), indicates that a potentially more optimal functional medicine approach would be to support the declining function of the endocrine (hypothalamus–pituitary–thyroid–adrenal–ovarian, HPTAO) axis, thereby creating hormonal “balance”.

With this novel mechanism of action of supporting the endocrine axis and regulating the endogenous production of hormones despite the absence of the introduction of exogenous hormones, it would seem that this would be an ideal first step as a treatment for many women. From that point forward, a clinician could assess if there is a need for additional exogenous hormone support and, if so, which hormones and to what degree, aligning with the current North American Menopausal Society (NAMS), Endocrine Society, and National Health Service (NHS) recommendations of personalizing the dose (often the lowest) and the duration of HRT (often the shortest) to the individual patient to maximize benefits and minimize risks [159,160,161]. Furthermore, this strategy offers an alternative for women who are over ten years beyond their menopausal transition, where there are heightened concerns about starting hormone therapy at an advanced age. It may also be a helpful, more natural approach for women who are looking for long-term hormone support throughout their life without side effects or risk or women for whom hormone therapy may not be an option or desired for various reasons (77% of women surveyed in one study were reluctant to do hormone therapy despite having symptoms) [162].

Apart from identifying phenotype (color), another point to consider is dose and concentration. Both Maca-GO^®^ and Maca-BG1.2^TM^ were noted as a concentration and an extract, respectively, versus Maca Power, which was undefined. Daily doses used in the abovementioned studies ranged between 300 mg and >3000 mg. Still, they did not define active ingredient parameters, except for Maca-GO^®^, which had glucosinolates as one of its standardized biomarkers and more information on sourcing described in the study methods (country, elevation, hypocotyl size, and manufacturing procedures). This additional information would enable greater accuracy in assessing and differentiating maca ingredients. Moreover, the genetic and/or cultural aspects of study participants could relate to the menopausal transition in a way that would result in maca products having differential effects.

In a published case report of a Caucasian female in her thirties taking a maca extract to improve energy and libido, there was evidence of increased plasma testosterone; however, no symptoms related to virilization [163]. Upon further investigation, the researchers noted that there was analytical inference in the immunoassay caused by maca supplementation [163]. Therefore, it is worth noting from a laboratory test point of view that there can be potential interference in the testosterone immunoassay in women taking a maca supplement. However, this is a single subject, and these results have not been reported elsewhere.

Of note, maca is also not a phytoestrogen with high concentrations of isoflavones like soybean and red clover [27,135,152], but it has been proposed to have progestin-like activity due to increases in uterine weight seen in a study with ovariectomized mice [164]. While it may not be a classical phytoestrogen, by nature of its potential ability to enhance endogenous E2 and P levels depending on the possible phenotypes used and the form it is in, its use would be contraindicated in those with a personal history of hormone-sensitive cancer. However, with several chemopreventative compounds present in some forms and phenotypes of maca, there will no doubt be additional investigation on this topic in future research [95].

### 7.3. Testicular and Prostate Health

Standardized formulations known as MacaPure M-01 and MacaPure M-02 derived from unspecified colors of maca *(L. meyenii)* root were shown to enhance sexual function in mice and rats, suggesting possible regulatory modulation of gonads through specific phytochemicals like macaenes, macamides, and certain unsaturated fatty acids and their amides [126]. It is of interest to note that increased libido in men due to taking maca may not be due to hormone changes. In a clinical study, men aged 21–56 years old were given two doses of gelatinized dehydrated maca root or placebo for twelve weeks. While sexual desire increased at eight and twelve weeks, the effect was not attributed to changes in serum T or E2 [50]. In another study in men of the same age range, there was no statistically significant difference in T, FSH, LH, prolactin, or E2 in men when given gelatinized maca root (no colors specified) or placebo in doses known to be used for aphrodisiac enhancement [165].

Men with mild asthenozoospermia and/or mild oligozoospermia given a non-specified color of maca (2 g/day) or placebo for 12 weeks presented statistically significant changes in sperm concentration [166]. However, the effects on sperm with this maca product are rather specific, considering the researchers found no significant differences in other parameters (sperm volume, mobility, and morphology) between the maca-supplemented and placebo groups [166]. Conversely, other studies using black maca in animal models have shown an improvement in parameters related to sperm, which may suggest that the color of maca may be relevant to men’s health [29,32,33,34,167,168]. A dried maca extract (of unspecified color) supplement given to non-Peruvian men for twelve weeks resulted in greater improvements in erectile dysfunction and well-being compared with those given a placebo [169].

Another important mechanism related to the endocrine system is its anti-inflammatory effect in the prostate gland through red maca’s impact on decreasing prostate weight (Figure 3) [51]. While the preponderance of data on prostate health advocates the use of red maca, there may be other colors of maca or combinations of maca that would also benefit specific physiological effects related to the function of this gland.

### 7.4. Thyroid Health

Subsequent clinical studies on Maca-GO^®^ also revealed some degree of positive impact on the thyroid hormone, triiodothyronine (T3), further suggesting that the hypothalamus–pituitary–thyroid–adrenal–gonad (HPTAG) axis may be affected through specific phenotypes and concentrations of maca, though this topic is less studied (Figure 3) [26,27].

## 8. Colors (Phenotypes)

Between thirteen to seventeen different colors (also known as “ecotypes” or phenotypes) of maca hypocotyls have been described [7,17,19,39,67,68,79,170] (Figure 4). In this paper, “colors” will be used interchangeably with “phenotypes”. The spectrum of colors of the maca hypocotyl that has been identified ranges from white to gray (lead) to black, as well as more colorful varieties such as red, red-white, red-yellow, white-red, white-lilac, white-purple, yellow, yellow-red, yellow-purple, purple (violet), purple-white, purple-gray, light gray, and light-gray-yellow [72,170,171]. Since it would be challenging to visually categorize each of these shades and mixtures of colors as there could be subjective interpretation, the main colors can be consolidated into purple (violet), red, black, white (“creamy white”), gray (also referred to as lead), and yellow [65,71,79]. Most published research features yellow, black, and red/purple varieties [5,72] (Figure 2). Pink has also been identified as one of the colors of maca [9,65,79], although no significant research corresponds with this phenotype. The color pigments are only found on the outer layer, with a white inner part; except for yellow, which is also yellow on the inside [65].

Meissner et al. have previously published research of the phytochemical characterization of the main phenotypes (yellow, black, red, and purple) [72]. Research has identified that select colors of maca have DNA variations (although still identifying as the same genus and species), display distribution patterns of active constituents, and, most importantly, can elicit different physiological responses in the body and even gender affinity [9,17,39,67,68,103]. For example, black maca has been shown to affect sperm production in animal studies; however, yellow has a moderate impact, and red maca has no effect [5]. Conversely, select published research indicates that red maca can reduce prostate size in animals, with induction by testosterone enanthate, but yellow maca has only mild effects, and black maca has none at all [5]. This color classification of function may result from the active constituents, sources, and the final preparation of maca, as documented by the research of Meissner et al., indicating that the same color of maca and size of hypocotyl can have a different analytical profile and concentration even when comparing maca from two different locations within Peru, namely Junín and Ancash [67,103]. Thus, it is reasonable to assume that there could be additional findings on prostate health using other types of maca or combinations of maca and why there sometimes appear to be conflicting results from the physiological effects of the same color of maca.

While there is variation in each color’s function, there are also certain documented similarities, such as the black, red, purple, gray, and yellow colors of *L. peruvianum,* all demonstrating acetylcholinesterase and butyrylcholinesterase inhibition (albeit to varying degrees) in in vitro studies [39], thereby presenting potential in the application for a therapeutic natural product to address memory enhancement.

An important conclusion from the research on maca is that not all colors have the same function and must be personalized to the intended outcome. A survey of the published literature on maca shows that distinct health benefits are observed when a specific color of maca is administered, as will be summarized in the following sections. However, as previously stated and requiring emphasis, most maca supplements available to the average consumer do not delineate the contents into these colors, making it problematic to choose a dietary supplement that will be helpful for a specific condition. This issue is further compounded by a plethora of reports in the literature not specifying the color of maca studied. As outlined in Table 2, of thirty published clinical trials on maca, only half (50%) had identifiable colors reported either in the study methods or in a previous citation on the maca product used in their publication. In the sections below highlighting each color of maca, only those studies that reported the color of maca used were included.

## 9. Black Maca

In one study looking at the polysaccharide, lipid, and essential oil components of three colors of maca (*L. meyenii*) hypocotyls from Yunnan, China, black maca was shown to be slightly higher (62.95%) in antioxidant activity (as assessed by scavenging % of DPPH) than yellow (54.75%) or red phenotypes (59.54%) [9]. Using spray-dried extracts, Zevallos-Concha et al. found that black maca had greater polyphenol content and antioxidant potential than red maca [172]. Its high polyphenol level and antioxidant potential, while also shared by red maca, may be responsible, in part, for some of its functions listed below, including its ability to increase physical endurance [172,173].

The overview of research suggests that black maca supports men’s fertility, spermatogenesis in animals [4,5,29,30,32,33,34,174,175], memory [4,37], and physical endurance in both men and women [4,49,173]. A systematic review by Lee et al. [35] investigated the evidence for maca supplementation in improving semen quality in both infertile and healthy men. It concluded that the evidence was unclear and too scant to support its use. While the authors noted the absence of studies, one of the issues in making a proper assessment was the absence of defining maca colors within the individual studies to examine whether there was a targeted effect of one type of maca over another, which might be anticipated with black maca based on animal data [35].

### 9.1. Preclinical Studies

#### 9.1.1. Reproductive Health

Concentrated black maca extracts have consistently been shown to impact spermatogenesis, and even preferentially, in some cases, compared with other colors of maca [30]. When three different extracts (methanolic extract, butanol fraction, aqueous fraction) of black maca and yellow maca were compared to control their effects on sperm parameters in adult mice, both black and yellow maca were found to increase daily sperm count differentially, depending on the administered format [34]. The methanolic extract of black and yellow maca had a superior effect than the butanolic or aqueous fractions [34]. Aqueous extracts from black and yellow maca, but not red maca, improved epididymal sperm count [29].

In a study investigating different fractions of black maca on spermatogenesis in rats, the researchers identified that the ethyl acetate fraction from the hydroalcoholic extract of black maca was most impactful, suggesting that compounds in that fraction had a therapeutic effect [168]. Furthermore, in a study with male rats, Gonzales et al. determined that an aqueous black maca extract affects the spermatogenic cycle as early as one day after starting treatment [33].

#### 9.1.2. Brain Health

It may be that antioxidant polyphenols assist in several of the benefits of black maca, such as improved learning and memory [37,38,164,176]. Compared with yellow and red maca, black maca led to the best response in a water-finding task in trained ovariectomized mice. However, all three phenotypes improved finding latency and the results of the forced swimming test [164]. Oral intake of a hydroalcoholic extract of black maca demonstrated a dose–response effect in a mouse model of memory impairment due to ethanol intake (20%) [38]. The researchers suggested that the ability to lessen the adverse effects of ethanol-induced memory impairment was due, in part, to the polyphenolic content (such as quercetin and anthocyanins) of the black maca extract [38]. Similarly, female ovariectomized mice performed better in memory and spatial learning tests with an aqueous extract of black maca compared with the control group given distilled water [37], as did male mice with scopolamine-induced memory impairment given either aqueous or hydroalcoholic extracts of black maca [176]. The black maca extract’s antioxidant content and acetylcholinesterase inhibitory activities were involved in this memory improvement [37,176]. Interestingly, an antidepressant effect of black, red, and yellow maca was documented [164]. The authors suggested that flavonoids such as quercetin and anthocyanins may contribute to the positive outcomes.

Finally, a recent study using a mouse model commonly used to study autistic spectrum disorder (ASD) identified that oral supplementation of an aqueous extract of black maca (Chinese source) helped with social interaction and memory impairment [177]. Based on these preliminary results, it may be possible to use maca in other studies to examine whether it could help reduce social deficits in individuals with ASD.

#### 9.1.3. Metabolic Health

Preclinical research would suggest that black maca extract may have applications in dysmetabolism [32,178]. Streptozotocin-induced diabetic mice given black maca extract presented reduced glucose levels [32]. Golden hamsters fed a high-fat, high-fructose diet were administered different doses (300, 600, and 1200 mg/kg) of an aqueous maca powder extract over 20 weeks [178]. Body fat, liver metabolite profiling, and metabolomics relative to energy and lipid metabolism were assessed [178]. While the food intake did not change with the black maca extract administration, there were no significant increases in body weight, liver weight, and fat weight, indicating that metabolism was accelerated [178]. Serum lipids (cholesterol, triglycerides, and LDL-cholesterol) and insulin decreased, and insulin sensitivity was enhanced in the animals on the higher doses (600 and 1200 mg/kg) of the black maca extract [178]. Metabolomic profiling indicated favorable metabolic changes in pathways related to glycolysis, tricarboxylic acid cycle activity, peroxisome proliferator-activated receptor (PPAR)-alpha signaling activation, and enhanced beta-oxidation [178].

#### 9.1.4. Bone Health

Another peripheral benefit of black maca (and red maca) is a protective action on bone structure, an effect not observed with yellow maca [179]. In one study using ovariectomized rats, standardized hydroalcoholic extracts of black and red maca were each effective in promoting bone integrity to the same level of animals given E2, but without increasing uterine weight, thereby demonstrating a lack of estrogenic activity [179]. Since the polyphenol amounts of the extracts used in this study differed between the red (higher content) and black maca (lesser content), there might be other mechanisms facilitating these bone-protective effects in the absence of being estrogenic. The authors proposed that maca may act through the hypothalamus–pituitary axis as one potential mechanism [179].

#### 9.1.5. Human Studies (See Table 3)

In a clinical trial, 44 elite athletes of different types (shooting, racket sports, swimming) took 2500 mg of 100% concentrated black maca extract twice daily for eight weeks [173]. The findings revealed decreased inflammation, improved mitochondrial biogenesis, and improved physical fitness and performance in some athletic groups [173]. Similarly, greater tolerance to high altitude was seen in healthy adults taking either 3 g of either black maca or red maca extracts for 12 weeks compared with placebo; however, each form of maca had slightly different outcomes [132]. Black maca positively affected hemoglobin levels and reduced glycemia [132]. In sixteen university-level racket athletes, a black maca powder supplement (purchased from Essoco, Seoul, Republic of Korea) taken twice daily (5 g total intake per day for four weeks) resulted in a statistically significant decrease in ammonia levels and an increase in muscle strength and endurance [180].

Due to the risk of testosterone replacement therapy and the need for alternatives, Korean researchers recruited 80 men over 40 years of age with clinical symptoms of androgen deficiency and low testosterone, randomizing them to either a placebo or a total of five grams of gelatinized black maca with standardized levels of *n*-benzyl-hexadecanamide between 115 and 175 µg/g, daily for twelve weeks [46]. Beneficial effects were seen in the maca-treated group, with significant improvement on questionnaires such as the Aging Males’ Symptoms (AMS) scale, Androgen Deficiency in the Aging Males, International Index of Erectile Function (IIEF), and International Prostate Symptom Score (IPSS), with no severe side effects; however, no statistically significant changes in serum total testosterone or free testosterone were noted between the placebo and maca groups [46].

#### 9.1.6. Clinical Observations

Over the last twenty-five years of researching maca and its use with thousands of women, the authors have observed consistent patterns of responses, indicating how using the appropriate phenotype for a specific health condition can not only be the difference between it being efficacious or not, but that in some cases, application of the incorrect phenotype with a particular condition can have adverse effects. Specifically, a worsening effect has been observed with maca products containing proportionally greater amounts of black maca in women with hormonally imbalanced conditions such as polycystic ovarian syndrome (PCOS) and relatively higher levels of estrogen [181].

**Table 3 nutrients-16-00530-t003:** Summary of published clinical trials using black maca, modified from [182].

Study Description	Details on Maca: Species, Product, Location Grown and Cultivation Methods, Form(s), Dose and Route	Overview of Study Results
8-week intervention trial [173] in 40 elite athletes: shooting athletes (SA), racket sports athletes (RSA), fin swimming athletes (FSA)	Species: *Lepidium meyenii* Product: Not stated Location/Cultivation: Not stated Form: 100% concentrated black maca extract Dose and Route: 2300 mg twice daily in the AM and PM	Evaluation of parameters using a digital grip dynamometer and a battery of physical exercises revealed significant increases after black maca supplementation in muscle endurance (*p* < 0.01 for SA group, *p* < 0.05 for RSA group), agility (*p* < 0.05 for SA and RSA groups), muscle strength (*p* < 0.01 for FSA group), flexibility (*p* < 0.01 for FSH group), and power (*p* < 0.05 for RSA group; *p* < 0.001 for FSA group. After taking black maca for eight weeks, significant differences in lactic acid (mg/dL) (*p* < 0.05) and total cholesterol (TC, mg/dL) were noted in the SA group (*p* < 0.01); TC (*p* < 0.01) and CRP (*p* < 0.01) in the RSA group; and lactic acid (*p* < 0.05), TC (*p* < 0.001), and CRP (*p* < 0.001) in the FSA group.
4-week study in 10 female athletes [183] Article is published in Korean; abstract only in English, so limited information is available	Species: Unknown Product: Unknown Location/Cultivation: Unknown Form: Unknown Dose and Route: 5 g of maca daily	After four weeks of training and taking black maca, there was a significant decrease in BMI, inflammatory markers, and muscle endurance.
4-week controlled clinical trial [180] in 16 university racket athletes with two groups given black maca: control and resistance exercise *(most of article is in Korean)*	Species: Not stated Product: Black maca was purchased from Essoco, Seoul, Republic of Korea. Location/Cultivation: Not stated Form: PowderDose and Route: 2.5 g twice daily, taken with water	Statistically significant decrease in ammonia levels (*p* < 0.05) and increase in muscle endurance and strength (*p* < 0.05).
12-week, double-blind, randomized, parallel-group, placebo-controlled clinical trial [46] at two hospitals in Korea with 80 men over 40 years old with Aging Males’ Symptoms (AMS) score >/=27; three subjects stopped the trial.	Species: *Lepidium meyenii* Product: Not stated Location/Cultivation: Not stated Form: Gelatinized powder Dose and Route: 1000 mg, two tablets *, three times daily, total of ~5 g per day, taken before food; each 1000 mg capsule contained 833 mg of maca (* *The authors describe the maca as being given in tablet and capsule form*.)	AMS, International Index of Erectile Function (IIEF), and International Prostate Symptom Score (IPSS) were significantly improved (*p* < 0.0001). Androgen Deficiency in the Aging Males was significantly reduced (*p* < 0.0001), showing an improvement in androgen deficiency symptoms. No changes in T or PSA levels, lipids, body weight, or waist circumference.
12-week randomized controlled trial with 45 men with mild erectile dysfunction (ED), 30–60 years old; [184] reported in [185]	Species: Unknown Product: Unknown Location/Cultivation: Peru Form: Two formats: (1) commercial product, gelatinized and dried, (2) fermented black maca Dose and Route: 5 g in capsules (used for both forms)	Both forms showed significant improvements compared to placebo (*p* < 0.05).
12-week double-blind placebo-controlled trial [132] in 175 men and women living in low (150 m above sea level, LA) and high altitudes (4340 m above sea level, HA); 18 subjects withdrew	Species: *Lepidium meyenii* Product: Enterprise A-1 del Perú, Industrial Comercial SAC prepared the finished product. Location/Cultivation: Peru, cultivation methods not stated Form: Spray-dried extract, standardized Dose and Route: 3 g daily	About 50% of participants reported an increase in sexual desire. Effects on mood, energy, and Chronic Mountain Symptom (CMS) scores were better using red maca compared to black maca and placebo. Effects on mood, energy, and CMS scores were better using red maca compared to black maca and placebo. In the red maca group at weeks 8 and 12, 80% of all participants reported increased mood and 90% reported increased energy. Higher quality of life was reported in both red and black maca groups (*p* < 0.05). Black maca reduced hemoglobin levels in HA participants. In HA, black maca reduced glucose levels in weeks 8 and 12 (*p* < 0.05), red maca reduced glucose in week 8 (*p* < 0.01). Systolic blood pressure was reduced in weeks 8 and 12 in the HA group using black maca (*p* < 0.01). Red maca reduced CMS scores in weeks 4 (*p* < 0.05), 8 (*p* < 0.01) and 12 (*p* < 0.01). Black maca reduced CMS scores in weeks 8 and 12 (*p* < 0.05).

## 10. Red Maca

There is limited research on red maca, including, most robustly, the research of Gonzales et al. and his colleagues on prostate health, followed by bone, skin, and detoxification support. Similar to the research on black maca, there are scant human clinical data.

### 10.1. Preclinical Studies

#### 10.1.1. Prostate Health

In animals, the traditional use of red maca, more than yellow or black maca, is helpful for prostate health by inhibiting benign prostatic hyperplasia (BPH) [4,5,41,42,43,44], a common issue worldwide effecting 50–90% of older-aged men [186]. While the exact mechanisms remain unknown, it is widely accepted that the levels of androgenic hormones and hormone receptors may play critical roles [186]. One of the standard treatments for BPH is finasteride, a pharmaceutical agent with 5-alpha-reductase inhibition activity; however, it is not without unwanted side effects [57]. As a result, there has been an investigation into other natural therapies such as maca, specifically red maca [187]. Even though red maca is therapeutic for prostate health, its effects on men’s reproduction, such as sperm count and sperm motility, are less substantial compared with yellow and black maca varieties [29].

One study in rats found that administering red maca and finasteride reduced inflammatory cell count in the prostate in distinct and different manners [51]. Finasteride treatment resulted in increased interleukin-4 (IL-4), whereas red maca led to increased interferon-gamma (IFN-gamma) [51]. Thus, there are two different pathways, but the same outcome of reducing inflammation by inhibiting tumor necrosis factor-alpha (TNF-alpha) [51]. This increase in IFN-gamma was also observed in a separate study on ovariectomized rats [188]. A clinical trial with BPH-induced mice given either red maca or finasteride for 21 days resulted in reduced prostate weight (1.59 times) in the red maca group [43].

Different red maca extracts have mixed results in their ability to impact prostate size and hormone activity. For example, one study found no difference in red maca’s effect on its ability to reduce prostate size in rats treated with testosterone enanthate (TE) when given as a hydroalcoholic or aqueous extract of red maca [45]. Benzyl glucosinolate was the marker compound related to dose-dependent efficacy [45], as previously indicated in another study [44]. Different doses of freeze-dried aqueous extract of red maca given to BPH-induced rats led to reduced prostate weight in a dose-dependent fashion, which was better than the effect of finasteride [41].

Conversely, another study tested the ability of several types of non-toxic red maca extracts to reduce the effects of TE-induced BPH in rats. The researchers identified that the *n*-butanol but not the aqueous fraction reduced prostate weights, similar to finasteride, with the mechanism related to upregulating the specific expression of estrogen receptor (ER)-beta while not changing the ER-alpha or androgen receptor (AR) [186]. Similarly, another study found AR stimulation with aqueous extracts of red maca in a prostate cancer cell line [189]. This specificity is relevant, considering that ER-alpha and ARs tend to be pro-inflammatory and enhance cellular growth and proliferation. In contrast, ER-beta has the opposite effect by balancing the activities of the other two receptors [186]. Therefore, from this study, it is conceivable that there are specific lipophilic compounds, perhaps beta-sitosterol or alkaloids, within red maca that target hormones and their receptor activity [186].

It is also worth mentioning that because there was such a significant amount of published data on red maca and prostate health in the scientific literature between 2005 and 2010, further investigations into other colors of maca for prostate health may have been deterred. Additionally, the location of maca cultivation could play a significant role in potential activities, with all the original research based on red maca from Junín in Peru. Therefore, knowing where the maca originates from and its color could be critical. It has been observed that, in some cases, primarily in prostate indications such as prostatitis and other prostate-related symptoms such as decreased urination, yellow maca may be counterproductive for prostate health [181]. While research over the past fifteen years has focused predominantly on red maca being the most effective, unpublished research from Meissner et al. indicates that a synergistic combination of black and red maca may be more effective. When combined with a proprietary supercritical extract of saw palmetto berries, these three actives had significant antioxidant, anti-inflammatory, and cytotoxic activities in laboratory assays [52]. It has been shown that the maximum desired synergistic effect of the three components in the plant mixture depends on the accurate individual selection of the three components and their precise proportions, as determined in laboratory bioassays [52]. This combination may be one of the potential treatments used in preventing prostate diseases and provide a novel, supportive therapy for prostate hyperplasia [52].

#### 10.1.2. Bone Health

Red maca has been shown to reverse the detrimental effects of bone loss in ovariectomized rats with increased trabecular bone and second lumbar vertebrae [179]. As hormone levels were not impacted, the authors suggest that maca acts as a selective estrogen receptor modulator (SERM) from the polyphenol content, enhancing osteoblast activity and reducing osteoclast resorption.

#### 10.1.3. Skin Health

The anti-inflammatory effects of red maca may have applicability for other conditions, such as skin healing. A study in mice found that the delay in wound healing seen at high altitudes, associated with heightened inflammation relative to sea level, was reduced by topical administration of red maca extract compared with the control treatment [147]. Both red and black maca extracts improved recovery from oxidative stress in ovariectomized rats subject to hyperbaric hypoxia [190]. Skin healing from exposure to ultraviolet B radiation was examined in three groups of mice with topical application of hydroalcoholic extracts containing either red, yellow, or black maca leaves [191]. All three extracts were photoprotective, preventing sunburn, and had antioxidant activity; however, each had distinct biological effects. Those administered the red maca leaf extract had reduced lipid peroxidation, as assessed by thiobarbituric acid-reactive substance (TBARS) production in the liver, consistent with the higher antioxidant activity observed with this extract relative to the other two [191]. Note that this research is on the leaf fraction rather than on the commonly used hypocotyl of the maca plant.

#### 10.1.4. Detoxification

Research on utilizing maca for reducing the effects of toxic exposures has been published, particularly in animal models of toxicity. One study in rats showed that a specific extract of red maca administered at different doses for four weeks led to significant, dose-dependent reductions in acrylamide-induced oxidative stress through decreases in malondialdehyde in erythrocytes, brain and liver and hepatic alanine aminotransferase (ALT) and aspartate aminotransferase (AST), and standard liver function tests [192].

#### 10.1.5. Brain Health

Limited evidence supports the use of red maca for depression and learning. One study using an aqueous extract of red maca led to anti-depressive effects in male rats and increased learning and memory in ovariectomized rats [36].

#### 10.1.6. Human Studies (See Table 4)

Few human clinical trials exist to support the benefits of red maca. In one study consisting of 175 subjects, the effect of placebo, black, or red maca spray-dried extracts at 3 g daily were tested for 12 weeks to examine any differences in their impacts on quality of life, chronic mountain sickness (CMS), or on blood glucose, blood pressure, and hemoglobin levels at low or high altitudes [132]. Both red and black maca extracts, which contained macamides and the glucosinolate glucotropaeolin, improved mood, sexual desire, energy, health, and lowered the CMS score. Although modest, these variables were more pronounced in the participants taking red maca [132]. Taking the black maca extract resulted in better hemoglobin levels for those at high altitudes. The researchers noted that gamma-aminobutyric acid (GABA), an anti-anxiety neurotransmitter, was higher in red maca than black maca, but macamides and fatty acids were higher in black maca.

**Table 4 nutrients-16-00530-t004:** Summary of published clinical trials using red maca, modified from [182].

Study Description	Details on Maca: Species, Product, Location Grown and Cultivation Methods, Form(s), Dose and Route	Overview of Study Results
12-week double-blind placebo-controlled trial [132] in 175 men and women living in low (150 m above sea level, LA) and high altitudes (4340 m above sea level, HA); 18 subjects withdrew	Product: Enterprise A-1 del Perú, Industrial Comercial S.A.C. prepared the finished product. Location/Cultivation: Peru, cultivation methods not stated Form: Spray-dried extract, standardized Dose and Route: 3 g daily	About 50% of participants reported an increase in sexual desire. Effects on mood, energy, and Chronic Mountain Symptom (CMS) scores were better using red maca compared to black maca and placebo. Effects on mood, energy, and CMS scores were better using red maca compared to black maca and placebo. In the red maca group at weeks 8 and 12, 80% of all participants reported increased mood and 90% reported increased energy. Higher quality of life was reported in both red and black maca groups (*p* < 0.05). Black maca reduced hemoglobin levels in HA participants. In HA, black maca reduced glucose levels in weeks 8 and 12 (*p* < 0.05), red maca reduced glucose in week 8 (*p* < 0.01). Systolic blood pressure was reduced in weeks 8 and 12 in the HA group using black maca (*p* < 0.01). Red maca reduced CMS scores in weeks 4 (*p* < 0.05), 8 (*p* < 0.01) and 12 (*p* < 0.01). Black maca reduced CMS scores in weeks 8 and 12 (*p* < 0.05).

## 11. Yellow Maca

Research exclusively on yellow maca is sparser than that of black and red maca. In one study by Meissner et al., it was reported that over 47% of the percentage distribution of Peruvian maca hypocotyls were identified as yellow maca [67] and, with copious availability, yellow maca is the least costly [193]. Even though it may have been a more common phenotype at one point in time, other environmental variables could result in a shift in the percentage distribution year over year. As noted above, there are indications from research findings that suggest it is unique in function and composition (genetic, nutritional, and phytochemical) from the other two phenotypes [103].

Nutritional differences include higher protein (equivalent to the red maca phenotype) and significantly higher quantities of a specific fatty acid, the trans fatty acid configuration for oleic acid, elaidic acid [67]. Regarding phytochemical content, Meissner et al. identified that yellow maca (*L. peruvianum*) had consistently lower levels of glucosinolates (comprised exclusively of a single glucosinolate, m-methoxy-glucotropaeolin) [103] relative to the black, purple, and red phenotypes [67]. With similar compounds as the aforementioned black and red maca, namely the glucosinolates and macamides, yellow maca also has some degree of antioxidant capacity, and, therefore, has shown a positive impact on reducing oxidative stress in one animal (stallion) study [194].

Additionally, as a dry powder, the yellow maca phenotype (*L. peruvianum*) was consistently higher in microbial contamination than these three phenotypes, with predominance for Gram-positive aerobic *Bacillus* strains [67]. Thus, future studies might be of interest to analyze the microbiome properties of each phenotype to detail any differences relative to their functional outcomes, considering the close contact of the hypocotyl with the soil environment throughout its growing trajectory and implications for storage conditions.

### 11.1. Preclinical Studies

#### 11.1.1. Musculoskeletal Health

With maca’s ability to confer better physical endurance, investigations have been made into its mechanisms in skeletal muscle. Applying different concentrations of yellow maca to skeletal muscle cells resulted in beneficial changes to muscle hypertrophy, differentiation, and maturation, with increases in phosphorylation of Akt and mTOR relative to the control group, indicating that yellow maca may be addressing some of the mechanisms related to sarcopenia [195]. The researchers touted that triterpenoid saponins found in maca may be responsible to some extent for these changes [195].

#### 11.1.2. Reproductive Health

With the association of maca and fertility, yellow maca has been tested with varying success for its ability to enhance reproductive measures in female and male animal studies. It has benefits for sperm production [29], but not to the same extent as black maca [30], and moderate to negligible effects on prostate health, where red maca is indicated [5,30,44]. One study tested yellow maca by administering an aqueous extract to adult female mice and found greater litter size than the control group, which received distilled water [196]. A yellow maca methanolic extract with LC-MS detection of 88 compounds was given to stallions based on body weight for 60 days, resulting in improved semen samples and better antioxidant measures in blood, as assessed through various assays [194].

Sanchez-Salazar and Gonzales studied a boiled aqueous extract of yellow maca using hypocotyls of different sizes and under varying pH conditions, administered orally or intraperitoneally to male mice [197]. The results indicated that each of these variables were relevant for the effects of the extract on sperm count, with the best outcomes obtained using larger-sized hypocotyls, lower pH, and oral administration [197]. Similarly, different outcomes on sperm count in adult mice were seen when using methanolic extracts of yellow maca compared with its butanolic and aqueous fractions, with the methanolic extract outperforming the two fractions [34]. Thus, this study illustrates the complexity of variables (e.g., concentration, dose, form, heat and water treatment, phenotype of maca, part of plant used, pH conditions, route of administration) implicated in the eventual efficacy of a maca product.

#### 11.1.3. Human Studies (See Table 5)

A randomized, double-blind, placebo-controlled trial in 50 infertile male adults, with half given yellow maca extract (a total of 2.8 g/day) or placebo (consisting of milled apple fiber and sucrose in a 3:2 ratio), for 16 weeks reported that sperm concentration was improved by 40% (compared with placebo at 76%), although the results were not statistically significant [193]. Of the hormones tested (LH, FSH, prolactin, E2, T, free T), the only substantial change was a 27% decrease in free T with the yellow maca extract after 16 weeks of use [193]. The authors of this study suggested that favorable changes in sperm quality could happen through a non-steroid hormone mechanism. Additionally, more extensive clinical trials are required to examine the effect of yellow maca further, including dose, administration, and other variables that may be consequential for reproductive health in men.

Conversely, yellow maca administration in a double-blind, randomized, placebo-controlled pilot study of 20 healthy men demonstrated improvements in sperm concentration and motility compared to the placebo [198]. It is a possibility that the inconsistent effects of yellow maca on fertility may be due to its preparation, size of the hypocotyl, type of extract (e.g., aqueous or methanolic extract), fractions from the extract (e.g., butanolic or aqueous), pH of the extract, and mode of administration [34,197].

**Table 5 nutrients-16-00530-t005:** Summary of published clinical trials using yellow maca, modified from [182].

Study Description	Details on Maca: Species, Product, Location Grown and Cultivation Methods, Form(s), Dose and Route	Overview of Study Results
16-week single-center, randomized, double-blind, placebo-controlled trial [193] in 50 infertile men 28–52 years old; 11 dropped out of the study for various reasons including goal achieved (pregnancy of partner), new medical diagnosis, adverse event, and unknown reasons.	Species: *Lepidium meyenii* Product: Provided by Andean Roots Ltd. Location/Cultivation: Peru, cultivation methods not stated Form: Gelatinized and powdered dried bulb, less than 7% humidity, prepared according to a previous study. Dose and Route: 2.8 g per day (7 capsules of 400 mg each)	In the yellow maca group, sperm count rose 15% (placebo 102%), sperm concentration by 40% (placebo 76%), and there was a significant decrease in free T (27%). Neither group showed a significant improvement in the sperm parameters measured.
12-week, double-blind, randomized, placebo-controlled pilot study [198] in 20 healthy adult men, 20–40 years old, 2 were excluded for oligo-zoospermia	Species: *Lepidium meyenii* Product: Andean Roots SRL Location/Cultivation: Cerro de Pasco region of Peru 4200–4500 m above sea level, no cultivation methods stated Form: Gelatinized and powdered Dose and Route: 1.75 g per day, 5 capsules containing 350 mg each	Increase in semen parameters after 12 weeks of maca use; increase in sperm count by 20%, sperm concentration by 14%, sperm motility by 14%, semen volume by 9%; morphology of sperm by 21%. No impact on hormone levels.

## 12. Purple (Violet) Maca

Of maca hypocotyls from Junín, Peru, purple maca has been reported to vary in percentage distribution, ranging between 6 and 29% [67]. Meissner et al. described five diverse maca phenotypes representing the color purple: purple-white, white-purple, purple-gray, yellow-purple, and purple [67]. It is conceivable that the purple and red phenotypes may be construed as the same color since they may present similarly, subject to the viewer’s interpretation; however, as Meissner et al.’s research demonstrates, red and purple phenotypes have distinctly different analytical profiles [67]. In a study comparing the nutritional composition of four Peruvian maca phenotypes (*L. peruvianum*), the purple variety was significantly higher in fat than the yellow, red, and black phenotypes [67]. The phytochemical content has not been extensively explored beyond glucosinolates and macamides [71]. Relative to other phenotypes, purple maca has been reported as either having modest amounts of glucosinolates (Chinese origin) [71] or the highest amounts (Peruvian origin) [72]. This difference is most likely dependent upon location and growing differences. Meissner et al. suggested that the higher glucosinolates in purple maca could result from a response to the concentrated UV exposure at higher altitudes [72]. In an analysis of Chinese maca, the purple maca phenotype had macamides similar to those of yellow and white maca [71].

Overall, purple maca has not been well studied for its efficacy. However, purplish pigments in plant foods such as fruits, flowers, and vegetables are the phytochemical class of water-soluble, biologically active flavonoids known as anthocyanins [199]. Anthocyanins have been reported to be potent antioxidants for heart, liver, brain, and kidney health [199,200]. Blue-purple foods (and beverages) such as blueberries, blackcurrant, purple sweet potato, and grapes (grapeseed, raisins, and grape juice) have been demonstrated to have cognitive and neuroprotective benefits [201,202,203,204,205,206,207,208,209,210]. Therefore, it would be of interest to research whether there are cognitive, nerve, and/or brain health benefits of the purple maca phenotype due to the potential anthocyanin content.

### Preclinical Studies

One animal study of diabetic rats [211] indicated several significant enhancements of antioxidant activity in red blood cells and the liver, specifically superoxide dismutase and catalase. Further, glutathione contents in plasma were also increased and glucose levels were decreased.

## 13. Gray Maca

The gray maca phenotype is a lesser-known form of the plant, at 2–5.4% of the distribution in Junín [67]. However, it has been analyzed for its chemical composition [39]. The findings revealed that the black, gray, and red phenotypes of *L. peruvianum* appear to have a similar composition of glucosinolates compared with the yellow and purple varieties [39]. A unique result for gray *L. peruvianum* was the higher amounts of indolyl glucosinolates, or glucosinolates derived from tryptophan [212], in contrast to the other colors tested [39]; however, the translation to health benefits remains speculative.

Pre-clinical and human studies have not been conducted on this color.

## 14. Combination of Colors

### 14.1. Maca-GO^®^

A formulation containing a combination of maca colors, black, yellow, purple, and red, known as Maca-GO^®^ has been extensively studied for its impact on hormone health and related body systems, such as bone markers, with two published animal studies [87,213] and four human clinical trials [24,25,26,27]. This maca formulation is described as a standardized, concentrated, bioavailable, organic, gelatinized (98.5–99.8% on the gelatinization index using a proprietary process) formulation, using a specific ratio of maca (*L. peruvianum*) phenotypes cultivated in Junín and processed using traditional drying methods [87,213]. It has known amounts of glucosinolates (73.36 mmol/kg dry weight) and phytosterols, as reported in [87].

#### 14.1.1. Preclinical Studies

Two published animal studies featured the use of Maca-GO^®^. In one study [87], male and female rats were given either 0.75 g/kg or 7.5 g/kg body weight of Maca-GO^®^ or control for 28 and 90 days. Based on biochemical parameters and histology, it was determined that Maca-GO^®^ was safe for short-term and extended use. There were differential effects in male and female rats and at varying durations. Notably, in the female rats on the higher dose in the longer trial, P was increased, while E2 maintained consistent levels; however, no changes were seen in FSH, LH, or TSH. There were also decreases in blood cortisol levels in the short- and long-term trials, along with reduced body weight, lowered plasma triglycerides, and even an increase in mineral (calcium and phosphorus) deposition in bone and muscle tissues.

A second animal study [213] using mice and ovariectomized rats determined the acute safety of higher doses of Maca-GO^®^ with an LD_50_ > 15 mg/kg body weight and its effects on hormone levels, lipids, iron, cognition, and mood status. As reported by Meissner et al., Maca-GO^®^ would be non-toxic and safe for use in humans at the equivalent of approximately 1 kg per 66 kg body weight per person [25]. Antidepressant and sedative effects were observed in ovariectomized rats only. Overall, hormone-balancing effects were seen in animals given Maca-GO^®^.

The results from these animal studies suggested that Maca-GO^®^ was safe and could be administered to humans.

#### 14.1.2. Human Studies (See Table 6)

Maca-GO^®^ has been clinically tested in perimenopausal and early postmenopausal women for its effects on hormones, cardiovascular parameters, bone density, and menopausal symptoms. Four placebo-controlled clinical trials have been published on this concentrated phenotype combination [24,25,26,27]. With deeper exploration into maca phenotypes and potential key active constituents, Maca-GO^®^ has been designed and tested clinically in specific populations, demonstrating statistically significant effects on hormones in perimenopausal (N = 18) and early postmenopausal (N = 177) women without the introduction of exogenous hormones.

In a four-month crossover, double-blind, placebo-controlled trial, perimenopausal women taking 2 g of Maca-GO^®^ for two months had statistically significant increases in E2 and P [27]. Similarly, in early postmenopausal women, oral intake of Maca-GO^®^ compared with placebo resulted in increases in E2 and P, along with decreases in FSH and LH, suggesting modulation of the hypothalamic–pituitary axis [25]. Further support of this mechanism was indicated in one of the clinical trials with early postmenopausal women in which those taking Maca-GO^®^ experienced statistically significant changes in other pituitary-regulated hormones, including cortisol, adrenocorticotropic hormone (ACTH), and triiodothyronine (T3), compared to placebo [26].

Modest changes in blood lipids (increased HDL-cholesterol, decreased LDL-cholesterol, and triglycerides) were seen with Maca-GO^®^ versus placebo. Forearm bone density scores were statistically increased after four months of use of Maca-GO^®^, while during this same duration, those on the placebo had measurable losses in bone density. These favorable changes in bone density were accompanied by increases in E2 (135%) and a decrease in FSH (55%) [26].

Significant reductions in menopausal symptoms, notably hot flushes and night sweats [24,25,26,27], were observed using Kupperman’s Menopausal Index and Greene’s Menopausal Score (*p* < 0.001). The stimulating effects on hormones and reduction in menopausal symptoms were further enhanced by extending the use of Maca-GO^®^ in early postmenopausal women to eight months [24]. A published case report [146] in a 32-year-old Caucasian female with vasomotor symptoms and mood issues following a hysterectomy and oophorectomy indicated that a personalized nutrition and lifestyle program incorporating Maca-GO^®^ resulted in significant improvements and resolution of symptoms associated with menopause.

While most data collected on Maca-GO^®^ have been in older-aged women related to hormone decreases and related symptomatology, there are two published case reports touting its use in combined nutraceutical protocols for young, premenstrual women [22,23]. In a 30-year-old Caucasian woman diagnosed with premenstrual syndrome and premenstrual dysphoric disorder, improvements were seen in premenstrual headaches, menstrual cramping, and flow and psychiatric symptoms within three months. It is essential to highlight that this research was conducted using 1000 mg per day of a formulation consisting of 75% Maca-GO^®^ and 25% of one specific concentrated, gelatinized maca phenotype [214] (reported in the publication as the commercial name, FemmenessencePRO^®^ Harmony), in addition to bio-identical progesterone and a magnesium supplement [22]. Phytochemicals within Maca-GO^®^, such as glucosinolates, may continue to support a woman on HRT related to the healthy metabolism of hormones, although this has not yet been verified through clinical research.

After four months of taking the same commercial maca product, *Medicago sativa*, and magnesium, a 39-year-old Caucasian woman experienced resolution of multiple premenstrual symptoms, reporting balanced moods, improved satiety, reduced cystic acne, and regulated bowel movements, in addition to complete cessation of menstrual pain, cramping, and lighter menstrual cycles [23].

### 14.2. MACAXS^TM^

#### Preclinical Study

Despite multiple studies suggesting that maca did not have an impact on testosterone levels, a combination of black, red, and yellow maca (MACAXS^TM^) in one animal study [215] was shown to increase testosterone levels after 42 days. The authors note that the increased duration of use may be a contributing factor to the favorable outcome in hormone concentration.

**Table 6 nutrients-16-00530-t006:** Summary of published clinical trials using a combination of maca colors, modified from [182].

Study Description	Details on Maca: Species, Product, Location Grown and Cultivation Methods, Form(s), Dose and Route	Overview of Study Results
Case report [23] of a 39-year-old Caucasian female with premenstrual syndrome (PMS), dysmenorrhea, and menorrhagia	Species: *Lepidium peruvianum* Product: FemmenessencePRO^®^ HARMONY (contains Maca-GO^®^) from Symphony Natural Health, Inc. Location/Cultivation: Not stated, however, previous research indicates Maca-GO^®^ is cultivated, harvested, dried on the plantation site in Junín, Peru [24]. Form: Not stated; however, previous research states Maca-GO^®^ is gelatinized and standardized [27]. Dose and Route: 500 mg twice daily as capsules, taken orally upon waking and mid-afternoon	In four months, the patient experienced a resolution of multiple PMS symptoms, reporting balanced moods, improved satiety, reduced cystic acne, and regulated bowel movements, as well as complete resolution of menstrual pain, cramping, and lighter menstrual cycles.
Case report [146] of a 32-year-old post-menopausal woman who had a hysterectomy and oophorectomy	Species: *Lepidium peruvianum* Product: FemmenessencePRO^®^ (contains Maca-GO^®^) from Symphony Natural Health, Inc. Location/Cultivation: Not stated, however, previous research indicates Maca-GO^®^ is cultivated, harvested, and dried on the plantation site in Junín, Peru [24]. Form: Not stated; however, previous research states Maca-GO^®^ is gelatinized and standardized [27]. Dose and Route: 1000 mg twice daily, capsules, taken orally upon waking and mid-afternoon	Resolution of hot flushes and anxiety in two months, improved mood and sleep, as measured by the Kupperman’s Menopausal Index (KMI).
Case report [22] of a 30-year-old Caucasian female diagnosed with premenstrual syndrome (PMS) and premenstrual dysphoric disorder (PMDD)	Species: *Lepidium peruvianum* Product: FemmenessencePRO^®^ HARMONY (contains Maca-GO^®^) from Symphony Natural Health, Inc. Location/Cultivation: Not stated; however, previous research indicates Maca-GO^®^ is cultivated, harvested, dried on the plantation site in Junín, Peru [24]. Form: Not stated; however, previous research states Maca-GO^®^ is gelatinized and standardized [27]. Dose and Route: 500 milligrams once daily to (or and) 1000 mg twice daily. Doses varied throughout care, and were taken orally upon waking and mid-afternoon.	Improvements in premenstrual headaches, fatigue, menstrual cramping and flow, and intense psychiatric symptoms were reported within three months of taking the maca supplement, bio-identical progesterone therapy, and magnesium. Additionally, LH reduced to normal levels, resulting in a normal FSH/LH ratio.
Double-blind, placebo-corrected clinical pilot study [24] in 20 Caucasian, healthy, early post-menopausal women, 45–62 years old; 8 subjects did not complete the trial. There were two trials with one lasting three months and the other nine months. Note: *The 3-month study included 1 month of placebo, followed by 2 months of Maca-GO^®^. The 9-month study included 1 month of placebo, followed by 8 months of Maca-GO^®^.*	Species: *Lepidium peruvianum* Product: Maca-GO^®^ Location/Cultivation: Junín, Peru (4200 and 4500 m above sea level); cultivated, harvested, and dried on the plantation siteof Junín, organic status Form: Gelatinized, standardized, no chemicals used, maca root powder in capsules Dose & Route: 1000 mg twice daily as capsules, oral intake 30 min before morning and evening meals	In 2 and 8 months, there were significant decreases in FSH (*p* < 0.05) and increases in LH (*p* < 0.05); At month 8, there were significant increases in E2 and P (*p* < 0.05); significant reduction in menopausal symptoms was noted in 2 and 8 months (*p* < 0.05).
4-month, double blind, crossover, randomized pilot trial [27]: 2 months of placebo and 2 months of Maca-GO^®^ in 20 Caucasian, healthy, menstruating women, 41–50 years old; 2 did not complete the study	Species: *Lepidium peruvianum* Product: Maca-GO^®^ Location/Cultivation: Junín, Peru (4200 and 4500 m above sea level); cultivated, harvested, and dried on the plantation site of Junín, organic status Form: Gelatinized, standardized, no chemicals used, maca root powder in capsules Dose & Route: 1000 mg twice daily as capsules, oral intake 30 min before morning and evening meals	At two months, 95% of women had improvements in menopausal symptoms as assessed by the Kupperman’s Menopausal index. After two months of taking Maca-GO^®^, significant reduction in body weight, systolic and diastolic blood pressure (*p* < 0.05), and increased HDL (*p* < 0.01) and iron (*p* < 0.05) levels were noted. Significant reductions in menopausal symptoms were reported: hot flushes, excessive sweating, interrupted sleep (all at *p* < 0.01), nervousness, depression, heart palpitation (all at *p* < 0.05); significant increases seen in E2, FSH, and P (all at *p* < 0.01); significant reduction in ACTH (*p* < 0.05).
Two trials [25] conducted at four clinics in 168 Caucasian, healthy, early menopausal women, 49–58 years old Trial 1: 3-month double-blind, randomized, coordinated multi-center, outpatient clinical study in 102 women (88 completed) Trial 2: 4-month double-blind, randomized, coordinated multi-center, outpatient clinical study in 66 women (40 completed)	Species: *Lepidium peruvianum* Product: Maca-GO^®^ Location/Cultivation: Junín, Peru (4200 and 4500 m above sea level); cultivated, harvested, and dried on the plantation site of Junín, organic status Form: Gelatinized, standardized, no chemicals used, maca root powder in capsules Dose & Route: 1000 mg twice daily as capsules, oral intake 30 min before morning and evening meals	Trial 1: Significant increase in E2 (*p* < 0.001), decrease in FSH (*p* < 0.05), significant increase in HDL levels (*p* < 0.05); significant decrease in menopausal symptoms in one month (*p* < 0.001) of treatment and further reduction in the second months of treatments, followed by an increase in symptoms when returning to placebo (*p* < 0.001). Trial 2: Significant decrease in FSH and LH (*p* < 0.05), significant increase in E2 (*p* < 0.05), significant reduction in menopausal symptoms (*p* < 0.001).
**Trial 1:** 4-month double-blind, randomized, outpatient, four months crossover design clinical trial (2 month of treatment and 2 months of placebo) [26] in 22 Caucasian early post-menopausal women, 49–58 years old **Trial 2:** Inclusion of Trial 1 plus a pilot bone density observation in 12 Caucasian early post-menopausal women, 49–58 years old	Species: *Lepidium peruvianum* Product: Maca-GO^®^ Location/Cultivation: Junín, Peru (4200 and 4500 m above sea level); cultivated, harvested, and dried on the plantation site of Junín, organic status Form: Gelatinized, standardized, no chemicals used, maca root powder in capsules Dose and Route: 1000 mg twice daily as capsules; oral intake 30 min before morning and evening meals	Trial 1: Significant decrease in BMI and LH levels (*p* < 0.05), significant increase in E2 (*p* < 0.05); significant reductions in T3, cortisol, ACTH (*p* < 0.05); significant increase in serum iron and plasma calcium levels (*p* < 0.05); significant decrease in menopausal symptoms (*p* < 0.001) Trial 2: The group on Maca-GO^®^ had an increase in forearm bone density in four months, whereas the placebo group had a reduction in bone density during the same time. Maca-GO^®^ group had an increase in E2 and a decrease in FSH.

## 15. Unspecified Colors of Maca

### MacaPure M-01 and M-02

#### Preclinical Study

The formulation known as MacaPure, formulated from unspecified ratios of maca phenotypes standardized to specific macaene and macamide lipidic fractions (M-01 and M-02) through an extraction process, was orally administered to male mice and rats to determine its effect on sexual behavior in the animals [126]. The findings indicated an immediate effect of increased mounting behavior in the male mice and an improvement in erectile function of the testes-removed rats, suggesting that these fractions of maca have an aphrodisiac effect [126]. Thus, this study indicates scientific support for the folk medicine uses of maca for sexual function enhancement.

Despite the compelling evidence that the colors of maca vary in their composition and impact on various health conditions, much of the published literature does not disclose the color(s) studied, creating a challenge in understanding the potential health benefits and highlighting the need for color-specific studies to be completed. A 2022 systematic review and meta-analysis revealed that almost half of the articles did not state the color(s) used [216]. A summary of published clinical trials which do not specify the colors of maca used are found in Table 7.

## 16. Overview of Clinical Findings on the Colors of Maca

### 16.1. Plant Terminology and Cultivation

A review of the clinical trials indicates no unified lexicon for maca plant parts, leading to issues in outcome replication. As noted, the maca parts used for the study are often not discussed in the research methods. If stated, there is sometimes confusion or even overlap in the use of words. For example, researchers noting they utilized the root or bulb may reference the tuber (hypocotyl). Therefore, better botanical distinction is required in future studies.

The cultivation site, deemed to be of great importance, is often not provided in the study methods. However, when the growing location is mentioned in clinical trials, Peru is the predominant country that supplies maca products. Other relevant aspects that need to be mentioned are the plant’s size and stage of growth.

### 16.2. Phytochemical Content

The studies presented herein would suggest that research is diverse in the phytochemicals that are analyzed, with some focusing on glucosinolates, others on macamides, macaenes, or phytosterols. Sometimes, there is just one compound in the class of compounds that is analyzed. Therefore, the research lacks a more well-rounded, complete profile of maca phytochemicals, particularly in the clinical trials. Furthermore, a variety of analytical methods are employed, which may not consistently deliver the same results for phytochemicals.

Of the maca color types that have been researched, a common finding among them is the presence of novel phytochemicals in combination, such as the distinct glucosinolates they contain, the unique fatty acid amides, and even the presence of plant sterols. This combination of plant actives may be responsible for their antioxidant, anti-inflammatory, and even endocrine-modulating properties. As discussed herein, the identification of the cultivation site, in addition to color, may be an essential variable to consider in the quality and quantity of phytochemicals due to the role of environmental factors. Also, an extract standardized to an active, such as benzyl glucosinolate (as seen with Maca-BG1.2^TM^ supplied by CPX PERU S.A.C. or the maca supplement provided by Kinos Inc. in Tokyo, Japan), may result in specific effects related to that marker compound.

### 16.3. Clinical Efficacy Related to Preparation and Administration

Most studies do not document details on the methods related to the preparation of the maca product, whether it has been dried, gelatinized, or extracted. Raw, dried maca can have significant variability in the temperatures and methods used, resulting in destroyed active compounds or the formation of compounds that do not naturally occur when dried in industrial drying machines. Gelatinization consists of three primary variables (water, temperature, pressure), and each manufacturer could use different combinations of each, either enhancing or destroying the active ingredients, ultimately impacting the bioavailability and/or absorption into the body. As stated in many studies where extracts were used, different forms of extracts were the difference between individual phenotypes being clinically efficacious or not.

The other feature of clinical efficacy is establishing a set dose for an indication. The subjects took maca products in various ways: once or twice daily, with or without food, with the total amount ranging between 300 mg and 5000 mg daily. In some cases, a maca supplement was given together with other nutritional interventions, thereby limiting the ability to separate the effect of the maca product compared with the other actives administered to the patient.

### 16.4. Clinical Efficacy Related to Color

Currently, there is insignificant clinical data to support using one color of maca for one specific clinical indication (Table 7). Based on the literature review, while there appear to be some themes, it is difficult to strategically delineate one select color of maca for one desired effect. There may be an overlap between the functions of the various colors. Additionally, not all colors have been compared in most studies, so while the research may reflect a color of maca predominating for a particular condition, it has not consistently been compared against another color to suggest its ability to outperform other colors. Thus, a desired clinical outcome would best be obtained through a known maca formulation clinically tested for the indication used. In other words, not all maca products are similar in their effects.

### 16.5. Safety of Maca for Human Consumption or Therapeutic Use

Maca has a history of safe use as a food source for thousands of years in Peru. A Peruvian may consume up to 100 g daily [8]. The clinical trials have reported little to no adverse events or safety issues by taking maca products for up to eight months. A few studies documented starch intolerance or gastrointestinal effects, but these were usually not limiting enough to discontinue participation in the study. Therefore, the preparation of maca involving drying and gelatinization may be essential for maca’s digestibility and even bioavailability of its phytochemicals.

Finally, the only other safety consideration would be one involving conditions that would be influenced negatively by altering hormonal status since maca products have been able to modulate the endocrine system. However, this is difficult to assess as there is evidence of maca products regulating hormone production, which could also be protective or supportive for those conditions.

### 16.6. Endocrine System Optimization

Based on the preclinical and clinical data, there is the suggestion that maca, of sufficient qualities and doses, can help with adaptive responses and may assist with the interregulation of endocrine gland signaling by modulating stress response, hormone levels, sexual function, and even the functionality of the glands themselves, such as with the prostate gland. Therefore, there are specific types or colors of maca, either alone or in combination, which may assist with supporting or even optimizing the hormonal axis signaling for improved markers for both women’s and men’s health or amplifying endogenous hormone levels. As a result, a well-defined maca product that has demonstrated these effects in clinical studies could be contraindicated for select populations of individuals, depending on personal history or family history of conditions like hormone-sensitive cancers.

## 17. Emerging Ideas for Future Research

### 17.1. Newer Clinical Research Studies

Preclinical studies on maca are relatively dated. The clinical trials reflect a limited number of newer (past five years) studies and case reports on maca supplements, whether for fitness optimization or women’s health. Overall, more research on maca is required for the different clinical indications, and it would be worthwhile to have a comparative analysis of the various colors of maca to examine whether one outperforms another for a particular condition or whether they are synergistic in combination. There are minimal quality clinical data to support the use of maca phenotypes, except for peri- and post-menopausal symptom reduction.

### 17.2. Phytochemicals of Interest

Maca contains unique phytochemicals like glucosinolates, macaenes, macamides, and even certain fatty acids and their amides, which may be of research interest in various clinical applications, such as cognition and cancer. Because of their botanical proximity within the *Brassicaeae* family, testing ratios of maca and cruciferous vegetables in vitro and in vivo through dietary means would help assess whether they are additive or synergistic in their activities. Cruciferous vegetables have gained significant notoriety over the years for human health due to their glucosinolate and sulforaphane content, with dietary supplements formulated to maximize those features. It is plausible that these two vegetable classes would be an optimum combination for hormone health, immune-inflammatory response modulation, and upregulation of cellular defenses against environmental toxicants, to name a few.

As discussed above, with the ability that some maca products have to modulate endogenous production of hormones through the HPATO axis, together with their potential to properly detoxify and eliminate hormones through metabolic biotransformation pathways (yet to be studied more explicitly), maca may be an optimal therapeutic option in a clinical protocol for hormone balance for more than one mechanism.

### 17.3. Effects on Neurotransmitters

Since maca has reported mood effects, it may be of interest to explore further the levels of neurotransmitter-active compounds identified in maca, such as tryptophan and GABA, to determine whether these agents could be impacting stress response through not just the endocrine system and effects on cortisol but also the neurotransmitters, particularly in post-partum depression and other indications.

### 17.4. Gut Microbiome Modulation

The overarching effects of maca, and even some of the colors of maca, could conceivably be at the level of the gastrointestinal tract via the regulation of neurotransmitters, immune-inflammatory response, and even the microbiome. As discussed, the different microbial contents of various colors of maca may impart changes to the gut microbiome. The maca microbiome may be influenced by soil quality, environmental exposures, and even the inherent phytochemicals they contain. Further, with application to human health, it would be interesting to investigate how the high carbohydrate content of maca and its other nutrients could shape the microbial populations within the gut and even result in secondary metabolites of interest.

### 17.5. Gaps in the Literature on Maca

Even though maca has been used in traditional medicine for hundreds of years, there remain several outstanding issues to reconcile for its implementation in the present day for health conditions. For example, most clinical studies do not report various characteristics of maca that would be needed to make assessments regarding its use in therapeutic settings, such as color, standardizing its parts, noting size and weight, identifying the geographic location where it was cultivated, detailing the system of post-harvest handling, and any processing procedures, along with potentially exploring the maca microbiome and soil microbiome in which the maca grows.

Scientific research has generally advanced the understanding of maca phenotypes or distinct variations in color, phytochemicals, and even DNA. However, more current clinical studies are needed for further insight as many publications are more than a decade old. Up to seventeen colors have been identified; however, only three colors (black, red, yellow) have been researched extensively in animal studies, with black maca predominating in clinical trials (six publications) as a single phenotype and black, yellow, and red/purple as a combined formula (seven publications). Unfortunately, most studies do not state the exact colors of maca used (Table 7).

Each of these colors of maca has known phytochemicals for specific physiological functions (Table 8). Black maca is effective in modulating adrenal response and stamina, red maca affects hormone receptors, with favorable outcomes for prostate health, and yellow maca has demonstrated improvements in musculoskeletal markers and fertility. Still, most of these studies have been with animals and have not been conducted extensively with humans. Purple and gray maca have scant evidence, limited to their nutritional and phytochemical content.

## 18. Summary

Maca has a long history and use as food and medicine in Peru, yet consumer interest has outpaced the extent of clinical research available. Overall, select maca phenotypes have preclinical data suggesting they would have merit in clinical conditions concerning reproductive health, hormone balance, menopausal symptoms, stress, endurance, bone health, brain health, metabolic health, prostate health, and even mood. More research is needed. Future studies should consider the complexity of maca: its color, the location of where it is grown, and the exact content of phytochemicals it contains, in addition to the form used, concentration, and dose, to help further deepen the scientific understanding of this food, herb, and medicine for applications in human health.

## Data Availability

Not applicable.

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
