# Peer review of "Not All Maca Is Created Equal: A Review of Colors, Nutrition, Phytochemicals, and Clinical Uses"

_nutrients, 2024, doi:10.3390/nu16040530_

Round 1
Reviewer 1 Report
Comments and Suggestions for Authors
There are few minor comments of format/typos
The major problem is the data representation, some tables can be deleted and incorporated to the text and the most important table in the review is of very low quality, too much text, too many explanations/wording on the tables and data, dosages of products, metabolites and nutrients of interest, etc. are missing.
The whole table can be splitted in different tables (if needed) acording to diseases or compound classes and the most important part is to reduce importantly the text included in the columns. The texts should be placed in the manuscript results section and discussions, the talbe is mising dosages, less rewording on objectives (e.g. "objectives", Physical activity, inflammatory markers", "subjects" - n=40 elite athletes? sex? age?; "intervention trial" how? what product? what brand? what component?)
"Results" in table columns - be more strict on data! "inflammatory maker IL-6", not useful. IL6 = result? ug/ml? or whatever unit reported, P value of significance, DATA!!!
Species, product, part of the plant used, extract, etc. ALL! in one column
Metabolites detected in samples? which type of sampled studied? muscle? blood? urine?
"Colour" - ok, 1 column, but please, include compounds for THAT colour!
anthocyanins? mg/g? any data?
In general, the manuscript is rewordy and too descriptive and the texts in the tables are TOO dense, MORE data, LESS texts in the tables!!!!
The tables should be usfeful and reference for readers, not MORE text!
Comments on the Quality of English Language
English language and re-wording of the texts could benefit from reduction and sharper texts and paragraphs, besides reviewing style and typos.
Author Response
There are few minor comments of format/typos
Thank you for this comment. Unfortunately, the manuscript was sent to you in a format that did not accurately reflect the submitted manuscript. Apologies for this discrepancy. The revised manuscript is being sent to you in a Word PDF document before any formatting occurs for the journal.
The major problem is the data representation, some tables can be deleted and incorporated to the text and the most important table in the review is of very low quality, too much text, too many explanations/wording on the tables and data, dosages of products, metabolites and nutrients of interest, etc. are missing.
The whole table can be splitted in different tables (if needed) acording to diseases or compound classes and the most important part is to reduce importantly the text included in the columns. The texts should be placed in the manuscript results section and discussions, the talbe is mising dosages, less rewording on objectives (e.g. "objectives", Physical activity, inflammatory markers", "subjects" - n=40 elite athletes? sex? age?; "intervention trial" how? what product? what brand? what component?)
"Results" in table columns - be more strict on data! "inflammatory maker IL-6", not useful. IL6 = result? ug/ml? or whatever unit reported, P value of significance, DATA!!!
Species, product, part of the plant used, extract, etc. ALL! in one column
Metabolites detected in samples? which type of sampled studied? muscle? blood? urine?
"Colour" - ok, 1 column, but please, include compounds for THAT colour!
anthocyanins? mg/g? any data?
In general, the manuscript is rewordy and too descriptive and the texts in the tables are TOO dense, MORE data, LESS texts in the tables!!!!
The tables should be usfeful and reference for readers, not MORE text!
Dear Reviewer, the comments about the tables have been used to rework and revise the manuscript, and I believe your suggestions have made the manuscript now more reader friendly. Indeed, you were correct and astute in your observations about the number of tables and how one of the tables (Table 5) was presented. Thus, in this revised version, you will find the following:
- Tables 1, 2, and 4 have been deleted.
- Table 5 has been completely reworked from a large table with numerous columns down to individual tables for each of the colors of maca, using your suggestions to include a summary of the maca product details in one column, and to fortify the results sections with data specifics and to include p-values whenever they were provided.
- I especially appreciated your comment about anthocyanins; unfortunately, these types of details were not included in the studies. In fact, part of the reason for putting these tables together is to show the paucity of information provided in these studies. I would hope that better, higher-quality studies are published on maca products because of this review article. Your point about phytonutrients is well taken. I would also hope that these details are included in future research. Thank you for expressing this point.

Reviewer 2 Report
Comments and Suggestions for Authors
Comments:
The used template must be renewed (for 2023, not 2021).
There is a lot of unformatted text.
All Tables must be rewritten according to the journal requirements. The presentation of a sum of different glucosinolates in mmol/kg dry weight in Table 4 is irrelevant. The used mmol /kg is relevant only for a content of one compound. Table 1 and 2 are unnecessary. Table 5 must be in landscape.
Author Response
The used template must be renewed (for 2023, not 2021).
Thank you for this comment. The Nutrients editor has formatted this article within the chosen template. I would ask for the editor to make this modification.
There is a lot of unformatted text.
Agree, the version that was sent to the reviewers had many formatting issues that were not presented as such upon submission. In this revised version, the text formatting issues should be fixed. Please see the revised manuscript as was submitted.
All Tables must be rewritten according to the journal requirements. The presentation of a sum of different glucosinolates in mmol/kg dry weight in Table 4 is irrelevant. The used mmol /kg is relevant only for a content of one compound. Table 1 and 2 are unnecessary. Table 5 must be in landscape.
In accordance with your excellent suggestion, Tables 1, 2, and 4 have been removed. As the second reviewer also requested modifications to Table 5, it has been reworked significantly to divide up the colors of maca in their specific tables. These tables have been condensed for easy readability.

Round 2
Reviewer 1 Report
Comments and Suggestions for Authors
Authors dealed with the comments/suggestions and the data representation has been greatly improved. The review merits publication.
Comments on the Quality of English Language
I am not a native-speaker myself, but I would recommend English revision for language and style to eliminate typos and improve writings.